# Federated Learning with Heterogeneous Differential Privacy

## Abstract

Federated learning (FL) takes a first step towards preserving privacy by training statistical models while keeping client data local. Models trained using FL may still indirectly leak private client information through model updates during training. Differential privacy (DP) can be employed on model updates to provide privacy guarantees within FL, typically at the cost of degraded accuracy of the final trained model. Both non-private FL and DP-FL can be solved using variants of the federated averaging (FEDAVG) algorithm. In this work, we consider a heterogeneous DP setup where clients may require varying degrees of privacy guarantees. First, we analyze the optimal solution to a simplified linear problem with (heterogeneous) DP in a Bayesian setup. We find that unlike the non-private setup, where the optimal solution for homogeneous data amounts to a single global solution for all clients learned through FEDAVG, the optimal solution for each client in this setup would be a personalized one even when data is homogeneous. We also analyze the privacy-utility tradeoff for this problem, where we characterize the gains obtained from the heterogeneous privacy where some clients opt for less stringent privacy guarantees. We propose a new algorithm for federated learning with heterogeneous DP, referred to as FEDHDP, which employs personalization and weighted averaging at the server using privacy choices by clients, to achieve the Bayes optimal solution on a class of linear problems for all clients. Through numerical experiments we show that FEDHDP provides up to 9.27% performance gain compared to the baseline DP-FL for the considered datasets where 5% of clients opt out of DP. Additionally, we show a gap in the average performance of local models between non-private and private clients of up to 3.49%, empirically illustrating that the baseline DP-FL might incur a large utility cost when not all clients require the stricter privacy guarantees.

## 1 Introduction

The abundance of data and advances in computation infrastructure have enabled the training of high-quality machine learning models. On the other hand, the data is distributed over many devices that are typically power-constrained and have limited computational capabilities. To reduce the amount of data transmission over networks and maintain the privacy of raw data, (McMahan et al., 2017) proposed the federated learning (FL) framework for training a central server-side model using decentralized data at clients. See the recent surveys (Kairouz et al., 2019; Li et al., 2020; Wang et al., 2021) for more.

Federated learning frameworks aim to train a global model iteratively and collaboratively using clients' data. During each round, the server has access to a select number of clients, each of whom has a local dataset. The server broadcasts the current model to such clients, who train the model by taking gradient steps using their local data on the model and return the gradient-based update back to the server. The server then aggregates the updates and produces the new global model for the next round. Several prior works on federated learning algorithms have been proposed in the literature to overcome various issues that arise in realistic federated learning setups, e.g., data heterogeneity (Konečný et al., 2016; Zhao et al., 2018; Corinzia et al., 2019; Hsu et al., 2019; Karimireddy et al., 2020; Reddi et al., 2020), and device dropout and communication cost (Li et al., 2018; Zhu & Jin, 2019; Wang et al., 2020; Al-Shedivat et al., 2020).

Despite the clients' data being kept on device in federated learning, the deployed model at the central server is still vulnerable to various privacy attacks, such as membership inference attacks (Shokri et al., 2017) and model inversion attacks (Fredrikson et al., 2015), among others. In order to mitigate such a critical issue, privacy-preserving variations of federated learning algorithms are proposed in the literature. One promising

approach to privacy-preserving FL utilizes differential privacy in order to provide the privacy guarantees. Differential privacy is a widely studied and accepted mathematical notion that describes privacy-preserving algorithms where the information leakage of private data is bounded. Differential privacy is defined as follows

**Definition 1** (differential privacy (DP) (Dwork et al., 2014))**.** *A randomized algorithm $A(\cdot)$, whose image is denoted as $\boldsymbol{O}$, is said to be $(\epsilon, \delta)$-DP if for any two adjacent inputs $\boldsymbol{D}$ and $\boldsymbol{D'}$ that differ in just one entry, and all subsets $O \subseteq \boldsymbol{O}$ the following relationship holds*

$$\Pr(A(\boldsymbol{D}) \in O) \leq e^\epsilon \Pr(A(\boldsymbol{D'}) \in O) + \delta. \tag{1}$$

The difference in the two adjacent inputs in the DP definition can be a single sample, a single client, or a single set of clients depending on the application, keeping in mind that the condition must hold for all adjacent inputs. In federated learning, instead of targeting privacy guarantees for individual samples of each client, it is common to consider a different differential privacy guarantee by having the adjacent datasets describe the case where we seek to provide privacy at the client-level data (McMahan et al., 2018), i.e., global DP. Moreover, heterogeneous differential privacy has been a topic of interest in the literature and has been considered in various works such as (Alaggan et al., 2016; Jorgensen et al., 2015) that aim at examining the problem from a theoretical point of view, and (Zhou et al., 2020; Ferrando et al., 2021; Liu et al., 2021; Amid et al., 2022) that study combining public and private datasets to improve the utility of the model. It is worth noting that using differential privacy in federated learning causes unavoidable degradation in performance. Several prior works utilized differential privacy to provide privacy guarantees for federated learning algorithms. For example, (Truex et al., 2020; Sun et al., 2020; Kim et al., 2021; Song et al., 2015) apply DP mechanisms at clients to ensure local DP guarantees, where clients have complete control over the amount of privacy they desire. On the other hand, (Geyer et al., 2017; McMahan et al., 2018; Andrew et al., 2019; Wei et al., 2020; Bietti et al., 2022) apply DP mechanisms at the server to ensure global DP guarantees for all clients. Applying DP typically causes some degradation in utility, i.e., the model's performance degrades as the privacy budget gets smaller (Alvim et al., 2011; Sankar et al., 2013; Makhdoumi et al., 2014; Calmon et al., 2015). These approaches to privacy-preserving federated learning use fixed privacy budgets for all clients, an approach that can be overly strict and cause unnecessary degradation in performance. A variation of DP setups is proposed in the literature, for scenarios with clients and a server, where a hybrid model is used by combining local DP with global DP and giving clients the option to opt in either (Avent et al., 2017; Beimel et al., 2019). A *blender* process is considered by Avent et al. (2017) for computing heavy hitters where some clients opt in local DP while the remaining opt in global DP. Some drawbacks of these works include their assumption of clients' data to be IID, as well as applying local DP which requires a large number of samples at clients. These two assumptions make applying such an approach in FL setups difficult due to the non-IID nature of clients' data in FL, and the relatively small number of samples generated by clients in FL which requires either increasing the variance of the added noise or relaxing the privacy leakage budget leading to either large degradation in performance or higher privacy leakage budgets.

Heterogeneity is a fundamental feature of federated learning, e.g., clients' datasets can no longer be considered IID in FL (Li et al., 2020). This introduces a challenging and critical problem that needs to be resolved to realize the full potential of privacy-preserving FL in realistic environments. One possible solution is based on model personalization, where clients learn personalized local models that performs better on their local data compared to the global model when heterogeneity exists. There are different approaches to personalization in the literature by introducing different modifications to FL algorithms, e.g., (Smith et al., 2017; Wang et al., 2019; Arivazhagan et al., 2019; Khodak et al., 2019; Mansour et al., 2020; Fallah et al., 2020; Deng et al., 2020; Dinh et al., 2020; Li et al., 2021). Another type of heterogeneity include systems heterogeneity where different devices have different capabilities, in terms of various characteristics such as connection, computational, and power capabilities (Li et al., 2018). Solutions to system heterogeneity include designing algorithms that can tolerate device dropout, reduce communication cost, or reduce computations cost (Caldas et al., 2018; Gu et al., 2021; Horvath et al., 2021; Li et al., 2018). In this work, we study heterogeneity along the privacy axis. We find that, similar to other notions of heterogeneity, addressing this problem in an optimal fashion requires curating personalized solutions for each client, which is different from the homogeneous non-private setup where one global model can be used to serve all clients.

**Organization & Our Contributions**

In this work, we develop a framework to study heterogeneity in privacy requirements in federated learning setups. More specifically, we consider a new setup for privacy-preserving federated learning where privacy parameters are no longer fixed across all clients. We show that existence of non-private clients who may choose to relax their privacy choices, even if they represent a small percentage of the overall population, can be leveraged to improve the performance of the global model as well as the personalized local models for all clients. Our contributions and the organization of the paper are as follows:

- In Section 2, we propose a heterogeneous setup for privacy in federated learning frameworks. The proposed setup considers heterogeneity in privacy choices of clients in FL. Instead of granting the same level of privacy for all clients, each client is given the option to choose their desired level of privacy. Moreover, we formally pose an optimization objective for solving the problem from a client's point of view.

- In Section 3, we theoretically study heterogeneous differential privacy in the simplified Bayesian setup of federated point estimation, introduced by Li et al. (2021), where clients are either private or non-private (i.e., two levels of privacy). We show that unlike the case of *non-private FL with homogeneous data*[1], where the Bayes optimal solution is a single global model that could be learnt via vanilla federated averaging, the optimal Bayes solution in differentially private FL requires personalization, even in the case of homogeneous DP. We also characterize the optimal degree of personalization based on the privacy requirements, degree of data heterogeneity, and other parameters (See Theorem 3). Further, we characterize the privacy-utility tradeoff observed at clients. Finally, we discuss extension to federated linear regression and with more than two privacy levels.

- In Section 4, we propose the federated learning with heterogeneous differential privacy algorithm, referred to as FEDHDP, for the heterogeneous privacy setup. The FEDHDP algorithm extends the Bayes optimal solution for federated point estimation to be applicable to more realistic scenarios.

- In Section 5, we provide experimental results of the FEDHDP algorithm using various synthetic and realistic federated datasets from TensorFlow Federated (TFF) (Google, 2019) using reasonable privacy parameters, where the privacy choices presented to clients are either private or non-private. Although the design guarantees of FEDHDP don't apply in these complex settings, we experimentally show that it provides significant gains compared to DP-FEDAVG algorithm (Andrew et al., 2019), and other stronger variants of it.

## 2 Privacy Guarantees within Federated Learning

In this section, we briefly describe the federated learning (FL) setup together with existing privacy guarantees. FL consists of a central server, who wishes to learn a model, and a set of clients, who cooperate with the server to learn a model while keeping their data on device. In particular, the central server coordinates the training of the model using the clients over multiple training rounds. The set of all clients, denoted by $\mathcal{C}$, contains all clients that wish to cooperate in training the model. Each client $c_j \in \mathcal{C}$ has a local loss $f_j(\cdot)$ and a local dataset denoted by $\boldsymbol{D}_j = \{\boldsymbol{d}_{j_1}, \boldsymbol{d}_{j_2}, ..., \boldsymbol{d}_{j_{n_j}}\}$, where $\boldsymbol{d}_{j_i}$ is the $i$-th sample at the $j$-th client.

During communication round $t$, the server sends the current model state, i.e., $\boldsymbol{\theta}^t$, to the set of available clients in that round, denoted by $\mathcal{C}^t$, who take multiple gradient steps on the model using their own local datasets to minimize their local loss functions $f_j(\cdot)$. The clients then return the updated model to the server who aggregates them, e.g., by taking the average, to produce the next model state $\boldsymbol{\theta}^{t+1}$. This general procedure describes a large class of learning global models with federated learning, such as federated averaging (FEDAVG) (McMahan et al., 2017).

To design privacy-preserving federated learning algorithms using differential privacy, certain modifications to the baseline federated averaging algorithm are required. In particular, the following modifications are introduced: clipping and noising. Considering client-level privacy, the averaging operation at the server is the target of such modifications. Suppose that clients are selected at each round from the population of all clients of size $N$, with a certain probability denoted by $q$. First, each client update is clipped to have a norm

---

[1]Homogeneous data refers to the case where the data for all clients is independent and identically distributed (IID).

at most $S$, then the average is computed followed by adding a Gaussian noise with mean zero and co-variance $\sigma^2 I = z^2(\frac{S}{qN})^2 I$. The variable $z$ is referred to as the noise multiplier, which dictates the achievable values of $(\epsilon, \delta)$-DP. Training the model through multiple rounds increases the amount of leaked information. Luckily, the moment accountant method in (Abadi et al., 2016) can be used to provide a tighter estimate of the resulting DP parameters $(\epsilon, \delta)$. This method achieves client-level differential privacy defined in Definition 1. It is worth noting that the noise can be added at the client side but needs to achieve the desired resulting noise variance in the output of the aggregator at the server, which is still the desired client-level DP.

Selecting the clipping threshold as well as the noise multiplier is essential to obtaining useful models with meaningful privacy guarantees. During training, the norm of updates can either increase or decrease; if the norm increases or decreases significantly compared to the clipping norm, the algorithm may slow down or diverge. Andrew et al. (2019) presented a solution to privately and adaptively update the clipping norm during each round of communication in federated learning based on the feedback from clients on whether or not their update norm exceeded the clipping norm. We consider this as the baseline for privacy-preserving federated learning algorithm and refer to it in the rest of the paper as DP-FEDAVG (Andrew et al., 2019). The case where no noise is added is the baseline for non-private federated learning algorithm, which is referred to simply as NON-PRIVATE.

One fundamental aspect of DP-FEDAVG is that it provides an equal level of privacy to *all* clients. This naturally arises given the assumption that all clients have similar behavior towards their own privacy in the federated learning setup. In other words, DP-FEDAVG implicitly assumes a homogeneity of the privacy level is required by all clients. This is in contrast to the heterogeneity feature of federated learning setups, where different clients have different data, capabilities, and objectives. Next we describe our proposed setup for federated learning with heterogeneous differential privacy.

**Proposed Setup: Heterogeneous Privacy within Federated Learning**

The proposed setup for federated learning with heterogeneous differential privacy is as follows. Prior to training, the server presents each client with a set of different privacy parameters $\mathcal{P} = \{(\epsilon_1, \delta_1), (\epsilon_2, \delta_2), ..., (\epsilon_l, \delta_l)\}$. Each client $c_i \in \mathcal{C}$ then makes their choice from the set of privacy parameters based on their desired level of privacy. The server then creates $l$ subsets of clients who share the same choice of privacy parameters, i.e., $\mathcal{C}_1, \mathcal{C}_2, ..., \mathcal{C}_l$, each with their corresponding privacy parameters. The server then coordinates the training of a global model through updates from clients while ensuring the privacy of each group of clients according to their privacy parameters is met.

We further examine what the server and clients agree upon at the beginning of training a federated learning model in terms of privacy to formally define the considered privacy measures. Each client $c_j$, whose dataset is denoted as $\boldsymbol{D}_j$ that is disjoint from all other clients, requires the server to apply some randomized algorithm $A_j(\cdot)$, whose image is denoted as $\boldsymbol{O}_j$, such that the following holds

$$\Pr(A_j(\boldsymbol{D}_j) \in O_j) \leq e^{\epsilon_j} \Pr(A_j(\boldsymbol{D}_e) \in O_j) + \delta_j, \tag{2}$$

where $\boldsymbol{D}_e$ is the empty dataset, and the relationship holds for all subsets $O_j \subseteq \boldsymbol{O}_j$. This achieves client-level privacy with parameters $(\epsilon_j, \delta_j)$ from client $c_j$'s point of view. Let us assume we have $N$ clients, each has their own privacy requirements for the server $(\epsilon_j, \delta_j)$ for $j \in [N]$, which should hold regardless the choices made by any other client. Now, let us have a randomized algorithm $A(\cdot)$, which denotes the composition of all $A_j(\cdot)$'s; then, the parallel composition property of differential privacy states that the algorithm $A(\cdot)$ is $(\epsilon_c, \delta_c)$-DP, which satisfies the following:

$$\Pr(A(\boldsymbol{D}) \in O) \leq e^{\epsilon_c} \Pr(A(\boldsymbol{D}') \in O) + \delta_c, \tag{3}$$

where $\boldsymbol{D}$ contains all datasets from all clients and $\boldsymbol{D}'$ contains datasets from all clients but one, $\boldsymbol{O}$ is the image of $A(\cdot)$, and the relationship holds for all neighboring datasets $\boldsymbol{D}$ and $\boldsymbol{D}'$ that differ by only one client and all $O \subseteq \boldsymbol{O}$. The parallel composition property of differential privacy states that the resulting $\epsilon_c = \max_i \epsilon_i$, and $\delta_c = \max_i \delta_i$. Next, considering our setup, let us have $l$ sets of private clients $\mathcal{C}_i$'s. Each client in the $i$-th set of clients requires $(\epsilon_{p_i}, \delta_{p_i})$-DP, and without loss of generality, assume that $\epsilon_{p_i} \geq \epsilon_{p_l}$ and $\delta_{p_i} \geq \delta_{p_l}$ $\forall i < l$. This is the case we consider in this paper, where we apply a randomized algorithm $A_{p_i}(\cdot)$, whose image is denoted as $\boldsymbol{O}_{p_i}$, to the dataset that includes all clients in the set $\mathcal{C}_i$ and the following holds

$$\Pr(A_{p_i}(\boldsymbol{D}_{p_i}) \in O_{p_i}) \leq e^{\epsilon_{p_i}} \Pr(A_{p_i}(\boldsymbol{D}'_{p_i}) \in O_{p_i}) + \delta_{p_i}, \tag{4}$$

where $\boldsymbol{D}_{p_i}$ contains all datasets from all clients in $\mathcal{C}_i$ and $\boldsymbol{D}'_{p_i}$ contains datasets from all clients in that subset except one, and the relationship holds for all neighboring datasets $\boldsymbol{D}_{p_i}$ and $\boldsymbol{D}'_{p_i}$ that differ by only one client and all $O_{p_i} \subseteq \boldsymbol{O}_{p_i}$.

Now, let us assume in the proposed heterogeneous differential privacy setup that each client in $\mathcal{C}_i$ requires $(\epsilon_{p_i}, \delta_{p_i})$-DP in the sense of (2). As a result, we can see that the only way for DP-FEDAVG to *guarantee* meeting the privacy requirement for the clients in $\mathcal{C}_l$ with $(\epsilon_{p_l}, \delta_{p_l})$ is to enforce $(\epsilon_{p_l}, \delta_{p_l})$-DP for *all* clients. In other words, DP-FEDAVG needs to be $(\epsilon_{p_l}, \delta_{p_l})$-DP, i.e., it needs to apply the strictest privacy parameters to all clients in the sense of (3). On the other hand, in our setup we can *guarantee* meeting the privacy requirements for each set of clients by ensuring an $(\epsilon_{p_i}, \delta_{p_i})$-DP for clients in $\mathcal{C}_i$, respectively, in the sense of (4). In other words, we need to only apply the appropriate DP algorithm with its appropriate metrics for each subset of clients to ensure the privacy metrics are met. This in turn results in our setup satisfying the corresponding privacy requirements needed by each set of clients, which are the main targets that need to be achieved in both algorithms from the clients' point of view in terms of their desired privacy levels.

Next, in terms of objectives in federated learning setups, the server's goal is to utilize the clients updates by averaging them to produce the next model state, and in our case, these updates are subject to specific differential privacy conditions. On the other hand, clients have a different objective when it comes to their performance measures. The clients' goal is to minimize their loss function given all other clients datasets including their own. However, since clients do not have access to other clients' raw data, a client desires to use the information from the differentially-private updates computed using the datasets by other clients as well as its own local update in order to reach a solution. Assume that the client $c_j$ observes all other clients DP-statistics of the datasets $\{\boldsymbol{\psi}_i : i \in [N]\backslash j\}$, which are the outputs of a randomized function that satisfies the privacy condition in (2), as well as its own non-DP dataset $\boldsymbol{D}_j$. Then the client's Bayes optimal solution is

$$\boldsymbol{\theta}_j^* = \arg\min_{\widehat{\boldsymbol{\theta}}_j} \left\{ \mathbb{E}_{\mathcal{D}_j}\left[ \ell_j(\widehat{\boldsymbol{\theta}}_j) \big| \{\boldsymbol{\psi}_i : i \in [N]\backslash j\}, \boldsymbol{D}_j \right] \right\}. \qquad \text{(Local Bayes objective)}$$

where $\ell_j(\cdot)$ is the loss function used to train a model for client $c_j$, and $\mathcal{D}_j$ is the true distribution of the dataset at client $c_j$. Notice that client $c_j$ has access to their own dataset $\boldsymbol{D}_j$ and DP-statistics of the other datasets $\{\widetilde{\boldsymbol{D}}_i : i \in [N]\backslash j\}$. From the point of view of client $c_j$, this objective is the Bayes optimal solution when observing all DP-statistics from other clients that are subject to their privacy constraints. It is important to note that (Local Bayes objective) is not suited to a federated optimization framework due to the fact that even individual updates from other clients are not available to the client to utilize, but rather their aggregation through a global model at the server. In practice, each client utilizes the global model state $\widehat{\boldsymbol{\theta}}$ to optimize the following:

$$\widehat{\boldsymbol{\theta}}_j^* = \arg\min_{\widehat{\boldsymbol{\theta}}_j} \left\{ \mathbb{E}_{\boldsymbol{D}_j}\left[ \ell_j(\widehat{\boldsymbol{\theta}}_j) \big| \widehat{\boldsymbol{\theta}}, \boldsymbol{D}_j \right] \right\}. \qquad \text{(Local personalized federated objective)}$$

We notice that this solution is a form of personalization in federated learning, where clients no longer deploy the global model locally by default, but rather utilize it to derive better local models that perform well on their own local dataset. In the remainder of this paper we will demonstrate this approach's ability to learn good (even optimal as we shall see in the next section) personalized local models compared to baseline private federated learning. Next, we will consider the proposed setup for a simplified federated problem known as the federated point estimation.

## 3 Analyzing Heterogeneous Differential Privacy in Simplified Settings

In this section, we provide some insights into the heterogeneous differential privacy problem in a simplified setup inspired by the one proposed by Li et al. (2021). Recall that in the federated learning setup, clients are interested in learning good models that perform best on their local datasets. Specifically, in the federated point estimation setup, clients are interested in learning Bayes optimal models in the sense of (Local Bayes objective). We show that in this simplified setup, the solution could be cast as a bi-level optimization problem, which can be solved as a personalized federated learning problem (Local personalized federated objective). We first start by considering the global estimation on the server and

show the proposed solution is Bayes optimal for the other clients' data. Then we consider local estimations for all sets of clients and show that the proposed solution is Bayes optimal for all clients when using appropriate values of the the respective hyperparameters. We further characterize the privacy-utility tradeoff gains in the Bayes optimal solution compared to variants of differentially private federated averaging (Andrew et al., 2019).

### 3.1 Analyzing Heterogeneous Differential Privacy in Federated Point Estimation

We consider a setting consisting of $N$ clients, each with $n_s$ data points, where the goal of each client is to estimate the mean of their data. Let us denote the quantity to be estimated at client $c_j$ as

$$\phi_j = \phi + p_j, \tag{5}$$

where $\phi$ is a global parameter, and $p_j \sim \mathcal{N}(0, \tau^2)$ is an inherent Gaussian-distributed hyperparameter that encompasses the non-IID nature in federated learning setups we are interested in. Increasing $\tau^2$ makes the data more uncorrelated at different clients, i.e., leading to increase in data heterogeneity. Note that at the extreme $\tau^2 \to \infty$ the clients' data points are independent of each other. On the other hand, setting $\tau^2 = 0$ denotes the case of IID clients, i.e., data homogeneity. In other words, in this case, all client data are generated from the same distribution. The observed samples at client $c_j$ are denoted by $\boldsymbol{x}_j = \{x_{j,1}, x_{j,2}, ..., x_{j,n_s}\}$, and

$$x_{j,i} = \phi_j + v_{j,i}, \tag{6}$$

where $v_{j,i} \sim \mathcal{N}(0, n_s \alpha^2)$ is the additive noise in the observations. The loss function at the client $c_j$ is

$$f_j(\phi) = \frac{1}{2}\left(\phi - \frac{1}{n_s}\sum_{i=1}^{n_s} x_{j,i}\right)^2. \tag{7}$$

Then minimizing $f_j(\phi)$ leads the client to have the estimate $\hat{\phi}_j = \frac{1}{n_s}\sum_{i=1}^{n_s} x_{j,i}$ whose variance is $\alpha^2 + \tau^2$. For the sake of simplicity and clarity of analysis, we model the additive noise only as an addition at the server side rather than the client side. This way, when the server aggregates the private clients' updates, the resulting noise variance for privacy is equivalent to the desired value by the server. It is worth that the notion of privacy here remains a client-level privacy despite the location of noise addition. We denote the updates sent to the server by client $c_j$ as

$$\psi_j = \hat{\phi}_j + l_j, \tag{8}$$

where $l_j \sim \mathcal{N}(0, \gamma_j^2)$ for client $c_j$. Note that $\gamma_j^2$ is related to the value of the noise multiplier $z^2$ at the server. We will refer to our solution and algorithm as FEDHDP in the remainder of the paper. The algorithm's pseudocode for federated point estimation is described in Algorithm 1 in Appendix B. In this setup, the server and clients goals are to minimize the Bayes risk (i.e., test error), defined as follows

$$\theta^* := \arg\min_{\hat{\theta}}\left\{\mathbb{E}\left[\frac{1}{2}\left(\phi - \hat{\theta}\right)^2 \middle| \psi_1, ..., \psi_N\right]\right\}. \tag{9}$$

$$\theta_j^* := \arg\min_{\hat{\theta}}\left\{\mathbb{E}\left[\frac{1}{2}\left(\phi_j - \hat{\theta}\right)^2 \middle| \{\psi_i : i \in [N] \setminus j\}, \hat{\phi}_j\right]\right\}, \tag{10}$$

where the first corresponds to the server's Bayes objective, given all clients updates, while the second is the client's Bayes objective, given its non-private estimation as well as other clients' private updates.

In the case of federated point estimation considered here, and more generally for federated linear regression considered in Appendix A, the server's goal is to find the following:

$$\hat{\theta}^* := \arg\min_{\hat{\theta}}\left\{\frac{1}{2}\left\|\sum_{i\in[N]} w_i \psi_i - \hat{\theta}\right\|_2^2\right\}, \tag{11}$$

where $w_i$'s are the weights assigned to clients' updates in the averaging process, while each client has a goal to find the minimizer of their local objective function, i.e.,

$$\widehat{\theta}_j^* := \arg\min_{\widehat{\theta}} \left\{ \frac{1}{2}\|\widehat{\theta} - \widehat{\phi}_j\|_2^2 + \frac{\lambda_j}{2}\|\widehat{\theta} - \widehat{\theta}^*\|_2^2 \right\}, \qquad \text{(Local FEDHDP objective)}$$

where $\lambda_j$ is a regularization parameter that controls the closeness of the personalized model towards the global model. Higher values of $\lambda_j$ steers the personalized model to the global model, while smaller values of $\lambda_j$ steers the personalized model towards the local model at the client. Notice that (Local FEDHDP objective) is a special case of the (Local personalized federated objective) where personalization is performed through a bi-level regularization. Next, we will discuss why we chose this case and show that the proposed FEDHDP solution converges to the Bayes optimal solution for the server as well as the clients.

### 3.1.1 Federated Point Estimation with Private and Non-Private Clients

In our discussion so far, we assumed that clients have multiple privacy levels to choose from. In realistic setups, clients are expected to be individuals who may not have complete awareness about what each parameter means in terms of their privacy. It is worth noting that interpreting the meaning of $\epsilon$ in differential privacy is hard in general (Dwork et al., 2019), and thus it is unrealistic to assume the average client can make their choice precisely. Therefore, the server needs to make a choice on how these parameters are presented to clients. A special case we consider extensively in this paper is the case with two classes of *private* and *non-private* clients. Clients who choose to be private are guaranteed a fixed $(\epsilon, \delta)$-DP, while clients who choose otherwise are not private. In fact, the approach to privacy from the server's point of view is to enable privacy by default for all clients and give each client the option to opt out of privacy if they desire to do so, which is a more practical solution because clients can make informed decisions about their privacy, and those who are not as familiar with privacy choices are kept private by default. The non-private choice can be suitable for different types of clients such as enthusiasts, beta testers, volunteers, and company employees, among others. Before we present our proposed solution, we will discuss two major baselines that we consider in this setup:

**DP-FedAvg** uses the differentially private federated averaging (Andrew et al., 2019) as the global model, where all clients are guaranteed the same level of privacy regardless of their preference.

**HDP-FedAvg** ensures privacy for the set of private clients similar to DP-FEDAVG, however, no noise is added to the non-private clients' updates. By design, HDP-FEDAVG is a stronger variant of DP-FEDAVG for a more fair comparison in a heterogeneous privacy setting.

We present our algorithm, FEDHDP: *federated learning with heterogeneous differential privacy*, specialized to federated point estimation with private and non-private clients in Algorithm 1. There are three important parameters introduced in the algorithm: $r$ (the ratio of the weight assigned to private clients to that of the non-private ones), $\gamma^2$ (the variance of the noise added for differential privacy), and $\lambda_j$ (a hyperparameter that controls the degree of personalization at client $c_j$). We shall see that for different choices of these parameters, Algorithm 1 recovers HDP-FEDAVG as well as DITTO (Li et al., 2021), and strictly generalizes them. In the remainder of Section 3.1, we analyze Algorithm 1, and derive the optimal values for $r$ and $\lambda_j$ in order to achieve the Bayes optimal solution for the federated point estimation problem.

---

**Algorithm 1** FEDHDP: Federated Learning with Heterogeneous Differential Privacy (Point Estimation)

---

*Inputs:* $\theta^0$, $\alpha^2$, $\tau^2$, $\gamma^2$, $\eta = 1$, $\{\lambda_j\}_{j \in [N]}$, $r$, $\rho_{\mathrm{np}}$, $N$.
*Outputs:* $\theta^*$, $\{\theta_j^*\}_{j \in [N]}$
**At server:**
**for** client $c_j$ in $\mathcal{C}^t$ **in parallel do**
   $\psi_j \leftarrow ClientUpdate(\theta^t, c_j)$
**end for**
$\theta^* \leftarrow \frac{1}{\rho_{\mathrm{np}}N + r(1-\rho_{\mathrm{np}})N} \sum_{c_i \in \mathcal{C}_{\mathrm{np}}} \psi_i$
   $+ \frac{r}{\rho_{\mathrm{np}}N + r(1-\rho_{\mathrm{np}})N} \sum_{c_i \in \mathcal{C}_{\mathrm{p}}} \psi_i$

**At client $c_j$:**
$ClientUpdate(\theta^0, c_j)$:
$\theta \leftarrow \theta^0$
$\theta_j \leftarrow \theta^0$
$\theta \leftarrow \theta - \eta(\theta - \frac{1}{n_s}\sum_{i=1}^{n_s} x_{j,i})$
$\theta_j^* \leftarrow \theta_j - \eta_j\big((\theta_j - \frac{1}{n_s}\sum_{i=1}^{n_s} x_{j,i}) + \lambda_j(\theta_j - \theta)\big)$
$\psi \leftarrow \theta + \mathbb{1}_{c_j \in \mathcal{C}_{\mathrm{p}}}\mathcal{N}(0, (1-\rho_{\mathrm{np}})N\gamma^2)$
return $\psi$ to server

---

| Symbol | Notation |
|---|---|
| $N$ | Number of clients |
| $\rho_{\mathrm{np}}$ | Fraction of non-private clients |
| $\mathcal{C}$ | Set of clients |
| $\mathcal{C}_{\mathrm{np}}$ | Subset of non-private clients |
| $\mathcal{C}_{\mathrm{p}}$ | Subset of private clients |
| $n_s$ | Number of samples at each client |
| $n_s\alpha^2$ | Observation noise variance at a client |
| $\tau^2$ | Degree of data heterogeneity (non-IIDness) at clients |
| $\gamma^2$ | Privacy noise variance at the server |
| $\lambda_j$ | Regularization parameter at client $c_j$ |
| $r$ | Ratio of private weights to non-private weights on the server |

Table 1: Summary of notation for federated point estimation setup.

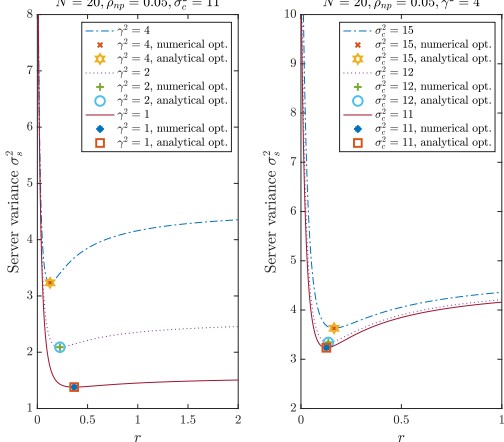

Figure 1: Server noise variance $\sigma_s^2$ vs the ratio hyperparameter $r$. (left) Trade-off for different $\gamma^2$, (right) trade-off for different $\sigma_c^2 := \alpha^2 + \tau^2$.

### 3.1.2 Optimal Global Estimate on the Server

The server's goal is to combine the updates received from clients such that the resulting noise variance is minimized, while ensuring the privacy of the set of private clients. Let us denote the sets of private and non-private clients as $\mathcal{C}_{\mathrm{p}}$ and $\mathcal{C}_{\mathrm{np}}$ respectively, and the fraction of clients with no privacy constraint (non-private clients) as $\rho_{\mathrm{np}}$. Also, let $\gamma_j^2 = 0$ for the non-private clients and $\gamma_j^2 = (1 - \rho_{\mathrm{np}})N\gamma^2$ for the private clients. To this end, we use Mahdavifar et al. (2017, Lemma 11) to find the optimal aggregator at the server. The server first computes the two intermediate average values for non-private and private clients as

$$\theta_{\mathrm{np}} = \frac{1}{\rho_{\mathrm{np}}N} \sum_{i \in \mathcal{C}_{\mathrm{np}}} \psi_i, \qquad \theta_{\mathrm{p}} = \frac{1}{(1 - \rho_{\mathrm{np}})N} \sum_{i \in \mathcal{C}_{\mathrm{p}}} \psi_i. \tag{12}$$

where $\theta_{\mathrm{np}} \sim \mathcal{N}(\phi, \frac{1}{\rho_{\mathrm{np}}N}(\alpha^2 + \tau^2))$, and $\theta_{\mathrm{p}} \sim \mathcal{N}(\phi, \frac{1}{(1-\rho_{\mathrm{np}})N}(\alpha^2 + \tau^2) + \gamma^2)$. Now, the server aims to combine such values to compute its estimation $\theta$ of the value of $\phi$ with the goal of minimizing the resulting estimation noise variance $\sigma_s^2$. If the resulting weighted average is expanded, it can be expressed as $\sum_{i=[N]} w_i\psi_i$. In this case, considering the weights used in the weighted average, let us denote the ratio of weights $w_i$'s dedicated for private clients to weights for non-private clients by $r = \frac{w_p}{w_{np}}$.

**Lemma 1** (Global estimate optimality). FEDHDP *from the server's point of view, with ratio* $r^* = \frac{\alpha^2 + \tau^2}{\alpha^2 + \tau^2 + (1 - \rho_{np})N\gamma^2}$, *is Bayes optimal (i.e.,* $\theta$ *converges to* $\theta^*$*) in the considered federated point estimation problem. Furthermore, the resulting variance is*

$$\sigma_{s,\mathrm{opt}}^2 = \frac{1}{N} \left[ \frac{(\alpha^2 + \tau^2)(\alpha^2 + \tau^2 + (1 - \rho_{np})N\gamma^2)}{\alpha^2 + \tau^2 + \rho_{np}(1 - \rho_{np})N\gamma^2} \right]. \tag{13}$$

The proof is relegated to the appendix to conserve space. Next, we show some simulation results for the server noise $\sigma_s^2$ against the ratio $r$ for different values of $\alpha^2 + \tau^2$ and $\gamma^2$ in the federated point estimation setup with $N$ clients and $\rho_{\mathrm{np}}$, fraction of non-private clients. The results are shown in Figure 1, and we can see that the optimal ratio $r^*$ in Lemma 1 minimizes the server variance as expected.

**Lemma 2** (Global model performance gap)**.** *The server-side estimation mean-square error gap between* FEDHDP *(optimal) and the baselines, HDP-*FEDAVG *and DP-*FEDAVG*, is as follows:*

$$\sigma^2_{s,hdp\text{-}fedavg} - \sigma^2_{s,opt} = \frac{\rho_{np}(1 - \rho_{np})^3 \gamma^4 N}{\alpha^2 + \tau^2 + \rho_{np}(1 - \rho_{np})\gamma^2 N} \geq 0, \tag{14}$$

$$\sigma^2_{s,dp\text{-}fedavg} - \sigma^2_{s,opt} = \frac{\rho_{np}(1 - \rho_{np})(\alpha^2 + \tau^2)\gamma^2 + \rho_{np}(1 - \rho_{np})^2 \gamma^4 N}{\alpha^2 + \tau^2 + \rho_{np}(1 - \rho_{np})\gamma^2 N} \geq 0, \tag{15}$$

$$\sigma^2_{s,dp\text{-}fedavg} - \sigma^2_{s,hdp\text{-}fedavg} = \frac{\rho_{np}(1 - \rho_{np})(\alpha^2 + \tau^2)\gamma^2 + \rho^2_{np}(1 - \rho_{np})^2 \gamma^4 N}{\alpha^2 + \tau^2 + \rho_{np}(1 - \rho_{np})\gamma^2 N} \geq 0. \tag{16}$$

Notice that if $\rho_{np} \to 0$ (homogeneous private clients) or $\rho_{np} \to 1$ (no private clients), the gap vanishes as expected. In other words, the benefit of FEDHDP on the server side is only applicable in the case of heterogeneous differential privacy. It can be observed that if the number of clients is large ($N \to \infty$), the gap approaches $(1 - \rho_{np})^2 \gamma^2$ and $(1 - \rho_{np})\gamma^2$ in (14) and (15), respectively. Notice that having this constant gap is in contrast to $\sigma^2_{s,opt}$ vanishing as $N \to \infty$. This is expected since the noise in the observation itself decreases as the number of clients increases and, hence, having the non-private clients alone would be sufficient to (perfectly) learn the optimal global model. Finally, if the noise $\gamma^2$ added for the privacy is large ($\gamma^2 \to \infty$), which corresponds to a small $\epsilon$, then the gap with optimality grows unbounded. In contrast, in this case, $\sigma^2_{s,opt}$ remains bounded, again because the optimal aggregation strategy would be to discard the private clients and to only aggregate the non-private updates.

### 3.1.3 Optimal Local Estimates on Clients

Now that we have characterized the optimal server-side aggregation, we consider optimal learning strategies at the client side. In particular, we show that FEDHDP achieves the Bayes optimal solution (Local Bayes objective) for local estimates at both the private as well as the non-private clients.

**Theorem 3** (Personalized local estimate optimality)**.** *Assuming using* FEDHDP *with ratio $r^*$ in Lemma 1, and using the values $\lambda^*_{np}$ for non-private clients and $\lambda^*_p$ for private clients stated below,* FEDHDP *is Bayes optimal (i.e., $\theta_j$ converges to $\theta^*_j$ for each client $j \in [N]$)*

$$\lambda^*_{np} = \frac{\alpha^2}{\tau^2}, \tag{17}$$

$$\lambda^*_p = \frac{\alpha^2(\tau^2 + \alpha^2) + \rho_{np}(1 - \rho_{np})\alpha^2 \gamma^2 N}{\tau^2(\tau^2 + \alpha^2) + (1 - \rho_{np})(\alpha^2 + \tau^2)\gamma^2 + (1 - \rho_{np})\rho_{np}\tau^2 \gamma^2 N}. \tag{18}$$

The proof follows from the properties of Gaussian random variables, (Mahdavifar et al., 2017, Lemma 11), together with algebraic derivations, which is relegated to the appendix. This result shows that FEDHDP recovers the (Local Bayes objective), which is the best one could hope for even if the client had access to differentially private versions of all other clients' updates without any constraints that arise in federated learning, such as just having access to the global model.

We also notice that the values of $\lambda^*$ are different for private and non-private clients. We recall that the derived expression for the personalization parameters for all clients consider the presence of data heterogeneity as well as privacy heterogeneity. In Table 2 we provide a few important special cases for both $\lambda^*_p$ and $\lambda^*_{np}$ for the considered federated point estimation problem.

Table 2: Special cases of FEDHDP in the federated point estimation.

| | No privacy ($\rho_{np} = 1$) | | Homogeneous privacy ($\rho_{np} = 0$) | |
|---|---|---|---|---|
| **Homogeneous data ($\tau^2 = 0$)** | FEDAVG | $\lambda^*_{np} = \infty$ | DP-FEDAVG+DITTO | $\lambda^*_p = \frac{1}{\gamma^2}$ |
| **Heterogeneous data ($\tau^2 > 0$)** | FEDAVG+DITTO | $\lambda^*_{np} = \frac{\alpha^2}{\tau^2}$ | DP-FEDAVG+DITTO | $\lambda^*_p = \frac{N\alpha^2}{N\tau^2 + (1 - \rho_{np})N\gamma^2}$ |

**Homogeneous data & No privacy:** In this case, the optimal FEDHDP algorithm recovers FE-DAVG (McMahan et al., 2017) with no personalization ($\lambda^*_{np} = \infty$). This is not surprising as this is exactly the setup for which the vanilla federated averaging was originally introduced.

**Heterogeneous data & No privacy:** In this case, the optimal FEDHDP algorithm recovers DITTO (Li et al., 2021). Again, this is not surprising as this is exactly the setup for which DITTO has been shown to be Bayes optimal.

**Homogeneous data & Homogeneous privacy:** In this case, the optimal FEDHDP algorithm recovers DP-FEDAVG (Andrew et al., 2019), however, with additional personalization using DITTO (Li et al., 2021). At first, this might be surprising as there is no data heterogeneity in this case, which is where DITTO would be needed. However, a closer look at this case reveals that the noise added due to differential privacy creates artificial data heterogeneity that needs to be dealt with using DITTO. In fact, as $\epsilon \to 0$, or equivalently, as $\gamma^2 \to \infty$, for the added noise for privacy, we observe that $\lambda_\mathrm{p}^* \to 0$ implying that the local learning becomes optimal. This is expected since, in this case, the data from other (private) clients is, roughly speaking, so noisy that it is best to rely solely on the personal data.

**Heterogeneous data & Homogeneous privacy:** In this case, the optimal FEDHDP algorithm again recovers DP-FEDAVG+DITTO. Similar to the homogeneous data case, with $\gamma^2 \to \infty$, we observe that $\lambda_\mathrm{p}^* \to 0$, i.e., the local learning becomes optimal.

**Remark:** Although the heterogeneous differential privacy problem is fundamentally different from robustness to data-poisoning attack (Li et al., 2021), its solution bears resemblance to a recently-proposed personalization scheme known as DITTO (Li et al., 2021). FEDHDP, in its general form, differs from DITTO in a number of major ways, and recovers it as a special case. First, the server-side aggregation in DITTO is the vanilla FEDAVG; however, in the proposed solution the server-side aggregation is no longer FEDAVG, but rather a new aggregation rule which utilizes the privacy choices made by clients. Second, DITTO, where the setup includes two sets of clients, i.e., benign and malicious, is designed for robustness against malicious clients; hence, the performance on malicious clients is not considered. On the other hand, the sets of clients in our proposed setup are the sets of clients with different levels of privacy; hence, measuring the performance across all sets of clients, i.e., clients with different privacy levels, is needed, and improving their performance is desired across all sets of clients. Third, the server in DITTO is unaware of the status of the clients, i.e., whether or not they are malicious; while in the proposed setup the server is aware of the privacy choices made by clients, and hence can give different weights to updates from private and non-private clients.

### 3.1.4 Privacy-Utility Tradeoff

Thus far, we have shown that for the problem of federated point estimation, the global estimate benefited greatly from the introduced setup of heterogeneous differential privacy. A better global estimate would enable better performance on clients' devices in the federated point estimation setup, even when no personalization is utilized. However, a question may arise on whether clients have a utility cost if they choose to remain private compared to the case where they opt out and become non-private.

To answer this question, we argue that opting out helps the server to produce a better global estimate, in addition to helping clients to produce better personalized local estimates. In other words, clients that opt out can produce better personalized local estimates compared to the ones that remain private. To illustrate the *motivation* of opting out for clients, we perform an experiment where we conduct the federated point estimation experiment for two scenarios. The first is the case where client $c_k$ remains private, and the second is the case where $c_k$ opts out of privacy and becomes non-private. For comparison, we provide the results of the experiments of FEDHDP with the optimal value $r^*$, as well as HDP-FEDAVG+DITTO. The results of these experiments are shown in Figure 2. We can see that if the client is non-private, they exhibit improvements in their estimates using the optimal value $\lambda^*$ for both algorithms, but the proposed FEDHDP with the optimal value $r^*$ greatly outperforms the one with vanilla FEDAVG. Additionally, in this problem, we can see that the optimal value of $\lambda_\mathrm{np}^*$ for non-private clients is always greater than or equal to the value $\lambda_\mathrm{p}^*$

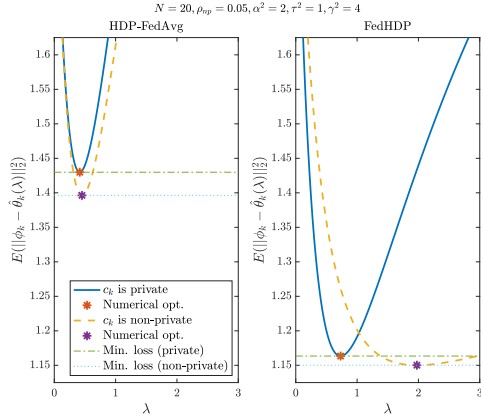

Figure 2: The effect of opting out on the personalized local model estimate as a function of $\lambda$ when employing (left) HDP-FEDAVG+DITTO and (right) FEDHDP.

for private clients, which is due to the value of $r$ being less than or equal to 1. In other words, non-private clients have more influence on the global estimate, and hence, encouraging the local estimate to get closer to the global estimate in (Local FEDHDP objective) is more meaningful compared to private clients. Furthermore, this experiment illustrates an important trade-off between privacy and utility for each client, where opting out of privacy improves performance, while maintaining privacy incurs degraded performance.

## 3.2 Extension to Federated Linear Regression with Multiple Privacy Levels

We consider the analysis for the federated point estimation with two privacy levels as a first step towards demonstrating the effectiveness of FEDHDP. We extend the analysis to the following setups in Appendix A:

- **Federated linear regression with two privacy levels:** In this extended setup, we have two subsets of clients $\mathcal{C}_1$ and $\mathcal{C}_2$, each having its own privacy requirements $\gamma_1^2$ and $\gamma_2^2$, respectively. We perform an analysis of the new setup, derive the expressions for the optimal hyperparameters under a diagonal covariance matrix assumption, which was also made by Li et al. (2021).

- **Federated linear regression beyond two privacy levels:** We also consider the case with more than two privacy levels in federated linear regression, provide an analysis of this setup, and show the solution to the optimal aggregator as well as the optimal regularization parameters for each set of clients. The solution would still be achieved by FEDHDP algorithm, however, now with different values of $\lambda_j$ for each set of client, and a server-side weighted averaging that depends on the individual values of noise variance $\gamma_i^2$ of each set of clients.

## 4 FedHDP: Federated Learning with Heterogeneous Differential Privacy

Now that we have been able to find a Bayes optimal solution in the simplified federated point estimation setup (and some extensions of it), we build upon the ingredients we used to build a general solution for federated learning with heterogeneous differential privacy. We formally present the FEDHDP algorithm and elaborate on its hyperparameters. The FEDHDP algorithm that is designed to take advantage of the aforementioned heterogeneous privacy setup is described in Algorithm 2. Similarly to the simplified setting, FEDHDP utilizes differential privacy with adaptive clipping, upweighting of less private clients on the server side, and a simple form of personalization.

---

**Algorithm 2** FEDHDP: Federated learning with heterogeneous DP (General Algorithm)

---

*Inputs:* model parameters $\boldsymbol{\theta}^0$, sensitivity $S^0$, learning rate $\eta$, personalized learning rate $\eta_p$, noise multipliers $\boldsymbol{z}, z_b$, quantile $\kappa$, and factor $\eta_b$.

*Outputs:* $\boldsymbol{\theta}^T, \{\boldsymbol{\theta_j}\}_{j\in[N]}$

**At server:**

**for** round $t = 0, 1, 2, ..., T-1$ **do**

  $\mathcal{C}^t \leftarrow$ Sample $N^t$ clients from $\mathcal{C}$

  **for** client $c_j$ in $\mathcal{C}^t$ **in parallel do**

    $\triangle\boldsymbol{\theta}_j^t, b_j^t \leftarrow ClientUpdate(\boldsymbol{\theta}^t, c_j, S^t)$

  **end for**

  **for** $j \in [l]$ **in parallel do**

    $N_j^t \leftarrow |\mathcal{C}_j^t|, \quad z_j^t \leftarrow z_j \frac{S^t}{N_j^t}$

    $\triangle\tilde{\boldsymbol{\theta}}_j^t \leftarrow \frac{1}{N_j^t} \sum_{c_i \in \mathcal{C}_j^t} \triangle\boldsymbol{\theta}_i^t + \mathcal{N}(\mathbf{0}, (z_j^t)^2\mathbf{I})$

  **end for**

  $\triangle\boldsymbol{\theta}^t \leftarrow \sum_{i\in[l]} w_i^t \triangle\tilde{\boldsymbol{\theta}}_i^t$

  $\boldsymbol{\theta}^{t+1} \leftarrow \boldsymbol{\theta}^t + \triangle\boldsymbol{\theta}^t$

  $S^{t+1} \leftarrow S^t e^{-\eta_b\left((\frac{1}{N^t}\sum_{i\in\mathcal{C}^t} b_i^t + \mathcal{N}(0, z_b^2 \frac{1}{N^t}^2)) - \kappa\right)}$

**end for**

**At client** $c_j$**:**

$ClientUpdate(\boldsymbol{\theta}^0, c_j, S)$:

$\boldsymbol{\theta} \leftarrow \boldsymbol{\theta}^0$

$\boldsymbol{\theta}_j \leftarrow \boldsymbol{\theta}^0$ (if not initialized)

$\mathcal{B} \leftarrow$ batch the client's data $\boldsymbol{D}_j$

**for** epoch $e = 1, 2, ..., E$ **do**

  **for** $B$ in $\mathcal{B}$ **do**

    $\boldsymbol{\theta} \leftarrow \boldsymbol{\theta} - \eta\nabla f_j(\boldsymbol{\theta}, B)$

    $\boldsymbol{\theta}_j \leftarrow \boldsymbol{\theta}_j - \eta_p(\nabla f_j(\boldsymbol{\theta}_j, B) + \lambda_j(\boldsymbol{\theta}_j - \boldsymbol{\theta}^0))$

  **end for**

**end for**

$\triangle\boldsymbol{\theta} \leftarrow \boldsymbol{\theta} - \boldsymbol{\theta}^0$

$b \leftarrow \mathbb{1}_{\|\triangle\boldsymbol{\theta}\|_2 \leq S}$

return $\text{Clip}(\triangle\boldsymbol{\theta}, S), b$ to server

$Clip(\boldsymbol{\theta}, S)$:

  return $\boldsymbol{\theta} \times \frac{S}{\max(\|\boldsymbol{\theta}\|_2, S)}$ to client

---

First, the notations for the variables used in the algorithm are introduced. The set of $N$ clients $\mathcal{C}$ is split into subsets containing clients grouped according to their desired privacy levels, denoted by $\mathcal{C}_1, \mathcal{C}_2, ..., \mathcal{C}_l$. Let the number of clients in the subset $\mathcal{C}_i$ be denoted by $N_i = |\mathcal{C}_i|$. The rest of the hyperparameters in the algorithm are as follows: the noise multipliers $\boldsymbol{z}, z_b$, the clipping sensitivity $S$, the learning rate at clients $\eta$, the personalized learning rate at clients $\eta_p$, quantile $\kappa$, and factor $\eta_b$. Also, the superscript $(\cdot)^t$ is used to denote a parameter during the $t$-th training round.

During round $t$ of training, no additional steps are required for the clients during the global model training. Clients train the received model using their local data followed by sending back their clipped updates $\Delta\boldsymbol{\theta}_j^t$ along with their clipping indicator $b_j^t$ to the server. The server collects the updates from clients and performs a *two-step aggregation process*. During the first step, the updates from the clients in each subset $\mathcal{C}_i$ are passed through a $(\epsilon_i, \delta_i)$ differentially private averaging function to produce $\triangle\tilde{\boldsymbol{\theta}}_i^t$. In the second step of the aggregation the outputs of the previous averaging functions are combined to produce the next iteration of the model. In this step, the server performs a weighted average of the outputs. The weights for this step are chosen based on the number of clients in each subset in that round, the privacy levels, as well as other parameters. This part resembles the weighted averaging considered in the aforementioned federated point estimation problem in Section 3.1.2. In general, the goal is to give more weight for updates from clients with less strict privacy requirement compared to the ones with stricter privacy requirements. The output of this step is $\triangle\boldsymbol{\theta}^t$, which is then added to the previous model state to produce the next model state.

To further elaborate on the averaging weights, let us reconsider the simple setup where we have only two subsets of clients, i.e., $\mathcal{C}_1$ and $\mathcal{C}_2$, with DP parameters $(\epsilon_1, \delta_1)$ and $(\epsilon_2, \delta_2)$, respectively. Also suppose that the second subset has stricter privacy requirements, i.e., $\epsilon_1 \geq \epsilon_2$ and $\delta_1 \geq \delta_2$ The weights $w_1^t$ and $w_2^t$ during round $t$ can be expressed as follows $w_1^t = \frac{N_1^t}{N_1^t + rN_2^t}$ and $w_2^t = \frac{rN_2^t}{N_1^t + rN_2^t}$. In general, we desire the value of the ratio $r$ be bounded as $0 \leq r \leq 1$ in FEDHDP to use the less-private clients' updates more meaningfully. The first factor to consider when choosing $r$ is related to the desired privacy budget, lower privacy budgets requires more noise to be added, leading to a lower value of $r$. This intuition was verified in the simplified setting in the previous section. Another factor that is more difficult to quantify is the heterogeneity between the less-private set of clients and the private set of clients. To illustrate this intuition we give the following example. Suppose that the model is being trained on the MNIST dataset where each client has samples of only one digit. Consider two different scenarios: the first is when each of the less-private clients have a digit drawn uniformly from all digits, and the second is when all of the less-private clients have the same digit. It can be argued that the ratio $r$, when every other hyperparameter is fixed, should be higher in the second scenario compared to the first; since contributions from the more-private clients are more significant to the overall model in the second scenario than the first. This will be experimentally verified in the experiments section presented later.

Then, clients need to train personalized models to be used locally. In the *personalization process*, each client simultaneously continues learning a local model when participating in a training round using the local dataset and the most recent version of the global model received during training and the appropriate value of $\lambda$. It is worth noting that the personalization step is similar in spirit to the personalized solution to the federated point estimation problem in Section 3.1.3. Furthermore, the server keeps track of the privacy loss due to the clients' participation in each round by utilizing the moment accountant method (Abadi et al., 2016) for each set of clients to provide them with tighter bounds on their privacy loss.

## 5   Experimental Evaluation

Thus far, we showed that FEDHDP achieves Bayes optimal performance on a class of linear problems. In this section, we present the results of a number of more realistic experiments to show the utility gain of the proposed FEDHDP algorithm with fine-tuned hyperparameters compared to the baseline DP-FEDAVG algorithm, where we additionally apply personalization on this baseline at clients using Ditto (Li et al., 2021) and refer to it as DP-FEDAVG+DITTO. Additionally, we compare the performance against another stronger baseline, HDP-FEDAVG+DITTO, which is a personalized variant of HDP-FEDAVG. Note that HDP-FEDAVG+DITTO can be viewed as a special case of FEDHDP with uniform averaging, i.e., $r = 1$, instead of a weighted averaging at the server. Note that the hyperparameters for the models, such as learning rate and others, are first tuned using the baseline NON-PRIVATE algorithm, then the same values of such

hyperparameters are applied in all the private algorithms. The experiments consider the case where two privacy levels are presented to each client to choose from, to be private or non-private. The experiments show that FEDHDP outperforms the baseline algorithms with the right choice of the hyperparameters $r, \lambda$ in terms of the global model accuracy, as well as in terms of the average personalized local model accuracy.

## 5.1 Setup

The experiments are conducted on multiple federated datasets, synthetic and realistic. The synthetic datasets are manually created to simulate extreme cases of data heterogeneity often exhibited in federated learning scenarios. The realistic federated datasets are from TFF (Google, 2019), where such datasets are assigned to clients according to some criteria. The synthetic dataset is referred to as the non-IID MNIST dataset, and the number of samples at a client is fixed across all clients. Each client is assigned samples randomly from the subsets of samples each with a single digit between $0-9$. A skewed version of the synthetic dataset is one where non-private clients are sampled from the clients who have the digit 7 in their data. In the non-IID MNIST dataset, we have $2,000$ clients and we randomly sample $5\%$ of them for training each round. The realistic federated datasets are the FMNIST and FEMNIST from TFF datasets. The FMNIST and FEMNIST datasets contain $3,383$ and $3,400$ clients, respectively, and we sample $\sim 3\%$ of them for training each round. TensorFlow Privacy (TFP) (Google, 2018) is used to compute the privacy loss, i.e., the values of $(\epsilon, \delta)$, incurred during the training phase. Refer to the appendix for an extended description of the used models and their parameters, as well as an extended version of the results.

**Remark:** It is worth noting that computing the optimal values of $r$, $\lambda_{\mathrm{np}}$, and $\lambda_{\mathrm{p}}$ for non-convex models such as neural networks is not an easy task. To resolve this issue in this section, we treat them as hyperparameters to be tuned. In practice, we cannot compute these parameters analytically, and hence we choose these parameters via grid search on the validation set. Note that tuning the regularization parameters $\lambda_{\mathrm{np}}$ and $\lambda_{\mathrm{p}}$ is done locally at clients and, hence, does not lead to any privacy loss. On the other hand, the ratio hyperparameter, $r$, needs tuning by the server, which results in some privacy loss. The recent work by Papernot & Steinke (2022) shows that tuning hyperparameters from non-private training runs incur significant privacy loss while such a hyperparameter tuning based on private runs may lead to negligible privacy loss. We leave a comprehensive investigation of this issue to future work.

## 5.2 Results

In this part, we provide the outcomes of the experiments on the datasets mentioned above. In these experiments, we provide results when $5\%$ of the total client population is non-private. Clients that opt out are picked randomly from the set of all clients but fixed for a fair comparison across all experiments. The exception for this assumption is for the skewed non-IID MNIST dataset, where clients that opt out are sampled from the clients who have the digit 7. All other hyperparameters are fixed. To evaluate the performance of each algorithm, we measure the following quantities for each dataset:

1. $Acc_{\mathrm{g}}$: the average test accuracy on the *server* test dataset using the global model.

2. $Acc_{\mathrm{g,p}}$, $Acc_{\mathrm{g,np}}$: the average test accuracy of all *private* and *non-private* clients using the global model on their local test datasets, respectively.

3. $Acc_{\mathrm{l,p}}$, $Acc_{\mathrm{l,np}}$: the average test accuracy of all *private* and *non-private* clients using their personalized local models on their local test datasets, respectively.

4. $\triangle_{\mathrm{g}}$, $\triangle_{\mathrm{l}}$: the gain in the average performance of *non-private* clients over the *private* ones using the global model and the personalized local models on their local test datasets, respectively; computed as $\triangle_{\mathrm{g}} = Acc_{\mathrm{g,np}} - Acc_{\mathrm{g,p}}$ and $\triangle_{\mathrm{l}} = Acc_{\mathrm{l,np}} - Acc_{\mathrm{l,p}}$.

A summary of the results, shown in Table 3 and Table 4, provides the best performance for each experiment along with their corresponding hyperparameters. More detailed results are shown in the appendix. If different values of the hyperparameters in FEDHDP yield two competing results, such as one with better global model performance at the server and one with better personalized models at the clients, we show both.

We can see from Tables 3 and 4 that FEDHDP allows the server to learn better global models while allowing clients to learn better personalized local models compared to the other baselines, i.e., DP-FEDAVG+DITTO

Table 3: Summary of the results of experiments on *synthetic datasets*: We compare the performance of the baseline algorithms against FEDHDP with tuned hyperparameters. The variance of the performance metric across clients is between parenthesis.

| nonIID MNIST dataset, $(3.6, 10^{-4})$-DP | | | | | | | | |
|---|---|---|---|---|---|---|---|---|
| Setup | | Global model | | | | Personalized local models | | |
| **Algorithm** | **hyperparameters** | $Acc_g\%$ | $Acc_{g,p}\%$ | $Acc_{g,np}\%$ | $\triangle_g\%$ | $Acc_{l,p}\%$ | $Acc_{l,np}\%$ | $\triangle_l\%$ |
| NON-PRIVATE DITTO | $\lambda_{np}=0.005$ | 93.8 | - | 93.75(0.13) | - | - | 99.98(0.001) | - |
| DP-FEDAVG+DITTO | $\lambda_p=0.005$ | 88.75 | 88.64(0.39) | - | - | 99.97(0.002) | - | - |
| HDP-FEDAVG+DITTO | $\lambda_p=\lambda_{np}=0.005$ | 87.71 | 87.55(0.42) | 88.35(0.34) | 0.8 | 99.97(0.001) | 99.93(0.001) | $-0.04$ |
| FEDHDP | $r=0.01$, $\lambda_p=\lambda_{np}=0.005$ | 92.48 | 92.43(0.30) | 93.30(0.21) | 0.88 | 99.94(0.001) | 99.94(0.001) | 0.0 |
| Skewed nonIID MNIST dataset, $(3.6, 10^{-4})$-DP | | | | | | | | |
| NON-PRIVATE DITTO | $\lambda_{np}=0.005$ | 93.67 | - | 93.62(0.15) | - | - | 99.98(0.001) | - |
| DP-FEDAVG+DITTO | $\lambda_p=0.005$ | 88.93 | 88.87(0.35) | - | - | 99.98(0.001) | - | - |
| HDP-FEDAVG+DITTO | $\lambda_p=\lambda_{np}=0.005$ | 88.25 | 88.05(0.39) | 89.98(0.05) | 1.93 | 99.97(0.001) | 99.85(0.001) | $-0.11$ |
| FEDHDP | $r=0.1$, $\lambda_p=\lambda_{np}=0.005$ | 90.36 | 89.96(0.37) | 97.45(0.01) | 7.49 | 99.97(0.001) | 99.76(0.003) | $-0.21$ |
| FEDHDP | $r=0.9$, $\lambda_p=\lambda_{np}=0.005$ | 87.96 | 87.69(0.56) | 92.97(0.04) | 5.28 | 99.98(0.001) | 99.96(0.001) | $-0.02$ |

Table 4: Summarized results of experiments on *realistic federated datasets*: We compare the performance of the baseline algorithms against FEDHDP with the hyperparameters that perform best. The variance of the performance metric across clients is between parenthesis.

| FMNIST dataset, $(0.6, 10^{-4})$-DP | | | | | | | | |
|---|---|---|---|---|---|---|---|---|
| Setup | | Global model | | | | Personalized local models | | |
| **Algorithm** | **hyperparameters** | $Acc_g\%$ | $Acc_{g,p}\%$ | $Acc_{g,np}\%$ | $\triangle_g\%$ | $Acc_{l,p}\%$ | $Acc_{l,np}\%$ | $\triangle_l\%$ |
| NON-PRIVATE DITTO | $\lambda_{np}=0.05$ | 89.65 | - | 89.35(1.68) | - | - | 94.53(0.59) | - |
| DP-FEDAVG+DITTO | $\lambda_p=0.05$ | 77.61 | 77.62(2.55) | - | - | 90.04(1.04) | - | - |
| HDP-FEDAVG+DITTO | $\lambda_p=\lambda_{np}=0.005$ | 75.87 | 75.77(2.84) | 74.41(2.8) | $-1.36$ | 90.45(1.02) | 92.32(0.8) | 1.87 |
| FEDHDP | $r=0.01$, $\lambda_p=0.05, \lambda_{np}=0.005$ | 86.88 | 85.36(1.89) | 90.02(1.28) | 4.66 | 93.76(0.68) | 95.94(0.41) | 2.18 |
| FEMNIST dataset, $(4.1, 10^{-4})$-DP | | | | | | | | |
| NON-PRIVATE DITTO | $\lambda_{np}=0.25$ | 81.66 | - | 81.79(1.38) | - | - | 84.46(0.89) | - |
| DP-FEDAVG+DITTO | $\lambda_p=0.05$ | 75.42 | 75.86(1.82) | - | - | 74.69(1.29) | - | - |
| HDP-FEDAVG+DITTO | $\lambda_p=\lambda_{np}=0.05$ | 75.12 | 75.87(1.65) | 78.59(1.58) | 2.72 | 74.67(1.34) | 75.95(1.12) | 1.28 |
| FEDHDP | $r=0.1$, $\lambda_p=\lambda_{np}=0.05$ | 76.52 | 77.91(1.67) | 83.9(1.27) | 5.99 | 77.9(1.22) | 79.15(0.99) | 1.25 |
| FEDHDP | $r=0.01$, $\lambda_p=\lambda_{np}=0.25$ | 74.86 | 77.31(2.18) | 86.73(0.98) | 9.42 | 81.19(1.02) | 84.68(0.78) | 3.49 |

as well as the FEDHDP with $r = 1$. For example, the gain due to FEDHDP compared to the DP-FEDAVG+DITTO in terms of global model performance is up to 9.27%. For personalized local models, the gain for clients due to FEDHDP compared to DP-FEDAVG+DITTO is up to 9.99%. Additionally, we can also see the cost in the average performance in personalized local models between clients who choose to opt out of privacy and clients who choose to remain private. This demonstrates the advantage of opting out, which provides clients with an incentive to opt out of differential privacy to improve their personalized local models, for example, non-private clients can gain up to 3.49% on average in terms of personalized local model performance compared to private clients. It is worth mentioning that opting out can also improve the global model's performance on clients' local data. We observe that there is up to 12.4% gain in the average performance of non-private clients in terms of the accuracy of the global model on the local data compared to the one of baseline DP-FEDAVG+DITTO. Similar trends can be observed for the other baseline.

## 6 Conclusion

In this paper, we considered a new aspect of heterogeneity in federated learning setups, namely, heterogeneous differential privacy. We proposed a new setup for privacy heterogeneity between clients where privacy levels are no longer fixed for all clients. In this setup, the clients choose their desired privacy levels according to their preferences and inform the server about the choice. We provided a formal treatment for the federated

point estimation problem and showed the optimality of the proposed solution on the central server as well as the personalized local models in such setup. Moreover, we have observed that personalization becomes necessary whenever data heterogeneity is present, or privacy is required, or both. We proposed a new algorithm called FEDHDP for the considered setup. In FEDHDP, the aim is to employ differential privacy to ensure the privacy level desired by each clients are met, and we proposed a two-step aggregation scheme at the server to improve the utility of the model. We also utilize personalization to improve the performance at clients. Finally, we provided a set of experiments on synthetic and realistic federated datasets considering a heterogeneous differential privacy setup. We showed that FEDHDP outperforms the baseline private FL algorithm in terms of the global model as well as the personalized local models performance, and showed an the cost of requiring stricter privacy parameters in such scenarios in terms of the gap in the average performance at clients.

## Acknowledgement

We are grateful to the anonymous TMLR reviewers whose thoughtful comments helped significantly with clarity of the presentation of this paper.

## Broader Impact & Limitations

In this paper, we investigated a heterogeneous privacy setup where different clients may have different levels of privacy protection guarantees, and in particular explored an extreme setup where some clients may opt out of privacy guarantees, to gain improvements in performance. However, the choice to loosen privacy requirements is heavily dependent on the client, the setting, and their valuation of their data. Moreover, since the algorithm orients the model towards the less private clients, it may introduce unfairness for the more private clients. Additionally, the server may have its own requirements during training, for example, a lower limit to the fraction of less private clients or vice versa, where the privacy choices may be overridden by the server. Overall, we believe that the interplay between all of these different societal aspects need to be carefully studied before the proposed mechanisms in this paper can be practically used.

We acknowledge that some of the assumptions in the theoretical study of the federated linear regression and federated point estimation setup are unrealistic, but similar assumptions have been made in prior work of (Li et al., 2021). For example, we neglected the effect of clipping, we assumed that all clients have the same number of samples, and assumed the covariance matrix is diagonal. On the other hand, for more complex models such as the ones used in the experiments, finding the best values of weights to be used in the aggregator at the server as well as the personalization parameters for each client is not straightforward, as some of the theoretical constructs in this paper are not estimable from data. Nevertheless, we believe that the theoretical studies in this paper can be used to build intuition about heterogeneous privacy setups and can be used as guiding principles for designing new algorithms.

Finally, the FEDHDP algorithm comes with two additional hyperparameters compared with DP-FEDAVG: $r$ (the weight ratio of private and non-private clients at the server), and $\lambda_j$ (the degree of personalization at client $c_j$). In this paper, we chose $r$ based on grid search, however that will naturally incur a loss of privacy that we did not carefully study. Having said that, recent work by Papernot & Steinke (2022) suggests that the privacy loss due to such hyperparameter tuning based on private training runs might be manageable but the exact interplay remains to be studied in future work.

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

## Appendix: Organization & Contents

We include additional discussions and results complementing the main paper in the appendix.

- Appendix A includes a discussion on federated linear regression and the optimality of FEDHDP along with some related simulation results.

- Appendix B includes proofs and additional results for the federated point estimation considered in the paper.

- Finally, we include additional information about the experimental setup along with extended versions of the results of each experiment in Appendix C.

# A    Theoretical Derivations for Federated Linear Regression

In this section, we consider applying the proposed algorithm FEDHDP on the simplified setup of federated linear regression problem, first introduced by Li et al. (2021). We utilize a similar setup as the federated point estimation problem, solve the optimization problem, and show the Bayes optimality of FEDHDP on the global model on the server, as well as the optimality of local models on clients.

## A.1    Federated Linear Regression Setup

Assume the number of samples per client is fixed and the the effect of clipping is negligible. Let us denote the total number of clients as $N$, of whom $N_1 = \rho_1 N$ are private with privacy level $p_1$ and $N_2 = (1 - \rho_1)N$ are private with privacy level $p_2$, where $p_2$ is stricter than $p_1$, i.e., more private. The subset of clients with privacy level $p_1$ is denoted by $\mathcal{C}_1$, while the subset of clients with privacy level $p_2$ is denoted by $\mathcal{C}_2$. Denote the number of samples held by each client as $m_s$, the samples at client $c_j$ as $\{\boldsymbol{F}_j, \boldsymbol{x}_j\}$, where $\boldsymbol{F}_j$ is the data features matrix, and $\boldsymbol{x}_j$ is the response vector. Let us assume that the features matrix $\boldsymbol{F}_j$ has the property of diagonal covariate covariance, i.e., $\boldsymbol{F}_j^T \boldsymbol{F}_j = n_s I_d$. Let us denote the relationship between $\boldsymbol{x}_j$ and $\boldsymbol{F}_j$ as

$$\boldsymbol{x}_j = \boldsymbol{F}_j \boldsymbol{\phi}_j + \boldsymbol{v}_j \tag{19}$$

where elements of the observations noise vector $\boldsymbol{v}_j$ are drawn independently from $\mathcal{N}(0, \beta^2)$, and $\boldsymbol{\phi}_j$ is the vector of length $d$ to be estimated. The vector $\boldsymbol{\phi}_j$ is described as

$$\boldsymbol{\phi}_j = \boldsymbol{\phi} + \boldsymbol{p}_j \tag{20}$$

where $\boldsymbol{p}_j \sim \mathcal{N}(0, \tau^2 I_d)$, and $\boldsymbol{\phi}$ is the vector to be estimated at the server. It is worth noting that $\tau^2$ is a measure of relatedness, as specified by Li et al. (2021), where larger values of $\tau^2$ reflect increasing data heterogeneity and vice versa. The local loss function at client $c_j$ is as follows

$$f_j(\boldsymbol{\phi}) = \frac{1}{m_s} \|\boldsymbol{F}_j \boldsymbol{\phi} - \boldsymbol{x}_j\|_2^2 \tag{21}$$

Local estimate of $\boldsymbol{\phi}_j$ at client $c_j$ that minimizes the loss function given $\boldsymbol{F}_j$ and $\boldsymbol{x}_j$ is denoted by $\widehat{\boldsymbol{\phi}}_j$ and is computed as

$$\widehat{\boldsymbol{\phi}}_j = \left(\boldsymbol{F}_j^T \boldsymbol{F}_j\right)^{-1} \boldsymbol{F}_j^T \boldsymbol{x}_j, \tag{22}$$

which is distributed as $\widehat{\boldsymbol{\phi}}_j \sim \mathcal{N}\left(\boldsymbol{\phi}_j, \beta^2 (\boldsymbol{F}_j^T \boldsymbol{F}_j)^{-1}\right)$. As assumed earlier, let $\boldsymbol{F}_j^T \boldsymbol{F}_j = n_s I_d$, then the loss function can be translated to

$$f_j(\boldsymbol{\phi}) = \frac{1}{2} \left\|\boldsymbol{\phi} - \frac{1}{n_s} \sum_{i=1}^{n_s} \boldsymbol{x}_{j,i}\right\|_2^2, \tag{23}$$

where $\boldsymbol{x}_{j,i}$'s are the noisy observations of the vector $\boldsymbol{\phi}_j$ at client $c_j$, which holds due to the assumption of the diagonal covariate covariance matrix. The updates sent to the server by client $c_j$ are as follows

$$\boldsymbol{\psi}_j = \widehat{\boldsymbol{\phi}}_j + \boldsymbol{l}_j \tag{24}$$

where $\boldsymbol{l}_j \sim \mathcal{N}(\boldsymbol{0}, N_1 \gamma_1^2 I_d)$ for private clients $c_j \in \mathcal{C}_1$ or $\boldsymbol{l}_j \sim \mathcal{N}(\boldsymbol{0}, N_2 \gamma_2^2 I_d)$ for private clients $c_j \in \mathcal{C}_2$. Note that as we mentioned in the paper, we still consider client-level privacy; however, we move the noise addition process from the server side to the client side. This is done such that when the server aggregates the private clients' updates in each subset the resulting privacy noise variance is equivalent to the desired value by the server, i.e., $\gamma_1^2$ and $\gamma_2^2$. This is done for simplicity and clarity of the discussion and proofs.

In this setup, the problem becomes a vector estimation problem and the goal at the server is to estimate the vector $\boldsymbol{\phi}$ given the updates from all clients, denoted by $\{\boldsymbol{\psi}_i : i \in [N]\}$ as

$$\boldsymbol{\theta}^* := \arg\min_{\widehat{\boldsymbol{\theta}}} \left\{ \mathbb{E}\left[ \frac{1}{2}\|\widehat{\boldsymbol{\theta}} - \boldsymbol{\phi}\|_2^2 \,\middle|\, \boldsymbol{\psi}_1, ..., \boldsymbol{\psi}_N \right] \right\}. \tag{25}$$

On the other hand, client $c_j$'s goal is to estimate the vector $\boldsymbol{\phi}_j$ given their local estimate $\widehat{\boldsymbol{\phi}}_j$ as well as the updates from all other clients $\{\boldsymbol{\psi}_i : i \in [N] \setminus j\}$ as

$$\boldsymbol{\theta}_j^* := \arg\min_{\widehat{\boldsymbol{\theta}}} \left\{ \mathbb{E}\left[ \frac{1}{2}\|\widehat{\boldsymbol{\theta}} - \boldsymbol{\phi}_j\|_2^2 \,\Big|\, \{\boldsymbol{\psi}_i : i \in [N] \setminus j\}, \hat{\boldsymbol{\phi}}_j \right] \right\}. \qquad \text{(Local Bayes objective)}$$

Now, considering the value of $\boldsymbol{\phi}$, the covariance matrix of client $c_j$'s update is denoted by $\Sigma_j$. The value of the covariance matrix can be expressed as follows

$$\Sigma_j = \begin{cases} \beta^2(\boldsymbol{F}_j^T\boldsymbol{F}_j)^{-1} + (\tau^2 + N_1\gamma_1^2)I_d, & \text{if } c_j \in \mathcal{C}_1 \\ \beta^2(\boldsymbol{F}_j^T\boldsymbol{F}_j)^{-1} + (\tau^2 + N_2\gamma_2^2)I_d, & \text{if } c_j \in \mathcal{C}_2 \end{cases} \tag{26}$$

We have $\boldsymbol{F}_j^T\boldsymbol{F}_j = n_s I_d$, and let $\frac{\beta^2}{n_s} = \alpha^2$, $\sigma_c^2 = \alpha^2 + \tau^2$, $\sigma_{p_1}^2 = \sigma_c^2 + N_1\gamma_1^2$, and $\sigma_{p_2}^2 = \sigma_c^2 + N_2\gamma_2^2$, then we have

$$\Sigma_j = \begin{cases} (\alpha^2 + \tau^2 + N_1\gamma_1^2)I_d, & \text{if } c_j \in \mathcal{C}_1 \\ (\alpha^2 + \tau^2 + N_2\gamma_2^2)I_d, & \text{if } c_j \in \mathcal{C}_2 \end{cases} \tag{27}$$

$$= \begin{cases} \sigma_{p_1}^2 I_d, & \text{if } c_j \in \mathcal{C}_1 \\ \sigma_{p_2}^2 I_d, & \text{if } c_j \in \mathcal{C}_2 \end{cases} \tag{28}$$

Next, we discuss the optimality of FEDHDP for the specified federated linear regression problem for the server's global model as well as the clients personalized local models.

## A.2 Global Estimate on The Server

In the considered federated setup, the server aims to find $\widehat{\boldsymbol{\theta}}^*$ described as follows

$$\widehat{\boldsymbol{\theta}}^* := \arg\min_{\widehat{\boldsymbol{\theta}}} \left\{ \frac{1}{2} \left\| \sum_{i \in [N]} w_i \boldsymbol{\psi}_i - \widehat{\boldsymbol{\theta}} \right\|_2^2 \right\}. \tag{29}$$

The server's goal, in general, is to combine the client updates such that the estimation error of $\boldsymbol{\phi}$ in (25) is minimized, i.e., the server aims to find the Bayes optimal solution described in (25). For the considered setup, the server aims to utilize the updates sent by clients, i.e., $\{\boldsymbol{\psi}_i : i \in [N]\}$, to estimate the vector $\boldsymbol{\phi}$. The estimate at the server is denoted by $\boldsymbol{\theta}$. Our goal in this part is to show that the solution to (29), which is the solution in FEDHDP algorithm converges to the Bayes optimal solution in (25). First, we state an important lemma that will be used throughout this section.

**Lemma 4** (Lemma 2 in Li et al. (2021)). *Let $\boldsymbol{\phi}$ be drawn from the non-informative uniform prior on $\mathbb{R}^d$. Also, let $\{\boldsymbol{\psi}_i : i \in [N]\}$ denote noisy observations of $\boldsymbol{\phi}$ with independent additive zero-mean independent Gaussian noise and corresponding covariance matrices $\{\Sigma_i : i \in [N]\}$. Let*

$$\Sigma_{\boldsymbol{\phi}} = \left( \sum_{i \in [N]} \Sigma_i^{-1} \right)^{-1}. \tag{30}$$

*Then, conditioned on $\{\boldsymbol{\psi}_i : i \in [N]\}$, we have*

$$\boldsymbol{\phi} = \Sigma_{\boldsymbol{\phi}} \sum_{i \in [N]} \Sigma_i^{-1}\boldsymbol{\psi}_i + \boldsymbol{p}_{\boldsymbol{\phi}}, \tag{31}$$

*where $\boldsymbol{p}_{\boldsymbol{\phi}} \sim \mathcal{N}(\boldsymbol{0}, \Sigma_{\boldsymbol{\phi}})$, which is independent of $\{\boldsymbol{\psi}_i : i \in [N]\}$.*

Next, we state the Bayes optimality of the solution at the server.

**Lemma 5** (Global estimate optimality). *The proposed solution, from the server's point of view, with weights $w_j$'s chosen below, is Bayes optimal in the considered federated linear regression problem.*

$$w_j = \begin{cases} \frac{1}{N_1 + N_2 r^*}, & \text{if } c_j \in \mathcal{C}_1 \\ \frac{r^*}{N_1 + N_2 r^*}, & \text{if } c_j \in \mathcal{C}_2 \end{cases} \tag{32}$$

*where*

$$r^* = \frac{\sigma_c^2 + N_1\gamma_1^2}{\sigma_c^2 + N_2\gamma_2^2}. \tag{33}$$

*Furthermore, the covariance of the estimation error is:*

$$\Sigma_{s,\text{opt}} = \frac{1}{N}\left[\frac{(\sigma_c^2 + N_1\gamma_1^2)(\sigma_c^2 + N_2\gamma_2^2)}{\sigma_c^2 + (1-\rho_1)N_1\gamma_1^2 + \rho_1 N_2\gamma_2^2}\right]I_d. \tag{34}$$

*Proof.* First, for the considered setup, Lemma 4 states that the optimal aggregator at the server is the weighted average of the client updates. The server observes the updates $\{\psi_i : i \in [N]\}$, which are noisy observations of $\phi$ with zero-mean Gaussian noise with corresponding covariance matrices $\{\Sigma_i : i \in [N]\}$. Then, the server computes its estimate $\theta$ of $\phi$ as

$$\theta = \Sigma_\theta \sum_{i\in[N]} \Sigma_i^{-1}\psi_i + p_\theta, \tag{35}$$

where $p_\theta \sim \mathcal{N}(\mathbf{0}, \Sigma_\theta)$ and

$$\Sigma_\theta = \left(\sum_{i\in[N]}\Sigma_i^{-1}\right)^{-1} = \left(N_1(\sigma_c^2 + N_1\gamma_1^2)^{-1}I_d + N_2(\sigma_c^2 + N_2\gamma_2^2)^{-1}I_d\right)^{-1} \tag{36}$$

$$= \frac{1}{N}\left[\frac{(\sigma_c^2 + N_1\gamma_1^2)(\sigma_c^2 + N_2\gamma_2^2)}{\sigma_c^2 + (1-\rho_1)N_1\gamma_1^2 + \rho_1 N_2\gamma_2^2}\right]I_d. \tag{37}$$

In FEDHDP with only two subsets of clients, we only have a single hyperparameter to manipulate server-side, which is the ratio $r$ that is the ratio of the weight dedicated for clients with higher privacy level to the one for clients with the lower privacy level. To achieve the same noise variance as in (37) we need to choose the ratio $r$ carefully. To this end, setting $r = \frac{\sigma_c^2 + N_1\gamma_1^2}{\sigma_c^2 + N_2\gamma_2^2}$ in FEDHDP results in additive noise variance in the estimate with zero mean and covariance matrix as follows

$$\Sigma_{s,opt} = \frac{1}{N}\left[\frac{(\sigma_c^2 + N_1\gamma_1^2)(\sigma_c^2 + N_2\gamma_2^2)}{\sigma_c^2 + (1-\rho_1)N_1\gamma_1^2 + \rho_1 N_2\gamma_2^2}\right]I_d. \tag{38}$$

Therefore, the weighted average of the updates using above weights, results in the solution being Bayes optimal, i.e., produces $\theta^*$. $\qquad\square$

## A.3 Personalized Local Estimates on Clients

As mentioned in Section 3, FEDHDP differs from DITTO in many ways. First, the global model aggregation is different, i.e., FEDAVG was employed in DITTO compared to the 2-step aggregator in FEDHDP. Second, in DITTO measuring the performance only considers benign clients, while in FEDHDP it is important to measure the performance of all subsets of clients, and enhancing it across all clients is desired. In this part, we focus on the personalization part for both sets of clients. The goal at clients is to find the Bayes optimal solution to the (Local Bayes objective). However, in the considered federated setup, clients don't have access to individual updates from other clients, but rather have the global estimate $\widehat{\theta}^*$. So, we have the local FEDHDP objective as

$$\widehat{\theta}_j^* := \arg\min_{\widehat{\theta}}\left\{\frac{1}{2}\|\widehat{\theta} - \widehat{\phi}_j\|_2^2 + \frac{\lambda}{2}\|\widehat{\theta} - \widehat{\theta}^*\|_2^2\right\}. \qquad\text{(Local FedHDP objective)}$$

First, we compute the Bayes optimal local estimate $\theta_j^*$ of $\phi_j$ for the local objective at client $c_j$. We consider client $c_j$, which can be either in $\mathcal{C}_1$ or $\mathcal{C}_2$, and compute their minimizer of (Local Bayes objective). In this case, the client is given all other clients' estimates $\{\psi_i : i \in [N] \setminus j\}$ and has their own local estimate $\hat{\phi}_j$. To this end, we utilize Lemma 4 to find the optimal estimate $\theta_j^*$. Given the updates by all other clients $\{\psi_i : i \in [N] \setminus j\}$, the client can compute the estimate $\phi^{\setminus j}$ of the value of $\phi$ as

$$\phi^{\setminus j} = \Sigma_{\phi^{\setminus j}}\left(\sum_{i=[N]\setminus j}\Sigma_i^{-1}\psi_i\right) + p_{\phi^{\setminus j}}, \tag{39}$$

where $\boldsymbol{p}_{\boldsymbol{\phi}\backslash j} \sim \mathcal{N}(\boldsymbol{0}, \Sigma_{\boldsymbol{\phi}\backslash j})$ and

$$\Sigma_{\boldsymbol{\phi}\backslash j} = \Big( \sum_{i=[N]\backslash j} \Sigma_i^{-1} \Big)^{-1}, \tag{40}$$

$$= \Big( m\frac{1}{\sigma_{p_1}^2}I_d + n\frac{1}{\sigma_{p_2}^2}I_d \Big)^{-1}, \tag{41}$$

$$= \frac{\sigma_{p_1}^2 \sigma_{p_2}^2}{n\sigma_{p_1}^2 + m\sigma_{p_2}^2}I_d, \tag{42}$$

where $n = N_2 - 1, m = N_1$ if $c_j \in \mathcal{C}_2$, or $n = N_2, m = N_1 - 1$ if $c_j \in \mathcal{C}_1$. Then, the client uses $\Sigma_{\boldsymbol{\phi}\backslash j}$ and $\hat{\boldsymbol{\phi}}_j$ to estimate $\boldsymbol{\theta}_j^*$ as

$$\boldsymbol{\theta}_j^* = \Sigma_{\boldsymbol{\theta}_j^*}\Big( (\Sigma_{\boldsymbol{\phi}\backslash j} + \tau^2 I_d)^{-1}\boldsymbol{\phi}^{\backslash j} + (\sigma_c^2 - \tau^2)^{-1}\hat{\boldsymbol{\phi}}_j \Big) + \boldsymbol{p}_{\boldsymbol{\theta}_j^*}, \tag{43}$$

$$= \Sigma_{\boldsymbol{\theta}_j^*}\Big( \Big(\frac{n\sigma_{p_1}^2 + m\sigma_{p_2}^2}{\sigma_{p_1}^2 \sigma_{p_2}^2 + \tau^2(n\sigma_{p_1}^2 + m\sigma_{p_2}^2)}\Big)\boldsymbol{\phi}^{\backslash j} + \frac{1}{\sigma_c^2 - \tau^2}\hat{\boldsymbol{\phi}}_j^* \Big) + \boldsymbol{p}_{\boldsymbol{\theta}_j}, \tag{44}$$

where $\boldsymbol{p}_{\boldsymbol{\theta}_j^*} \sim \mathcal{N}(\boldsymbol{0}, \Sigma_{\boldsymbol{\theta}_j^*})$ and

$$\Sigma_{\boldsymbol{\theta}_j^*} = \Big( \Big(\frac{\sigma_{p_1}^2 \sigma_{p_2}^2 + \tau^2(n\sigma_{p_1}^2 + m\sigma_{p_2}^2)}{n\sigma_{p_1}^2 + m\sigma_{p_2}^2}I_d\Big)^{-1} + ((\sigma_c^2 - \tau^2)I_d)^{-1} \Big)^{-1}, \tag{45}$$

$$= \frac{(\sigma_c^2 - \tau^2)\big(\sigma_{p_1}^2 \sigma_{p_2}^2 + \tau^2(n\sigma_{p_1}^2 + m\sigma_{p_2}^2)\big)}{\sigma_c^2\big(n\sigma_{p_1}^2 + m\sigma_{p_2}^2\big) + \sigma_{p_1}^2 \sigma_{p_2}^2}I_d. \tag{46}$$

We expand (44) as

$$\boldsymbol{\theta}_j^* = \frac{\sigma_{p_1}^2 \sigma_{p_2}^2 + \tau^2(n\sigma_{p_1}^2 + m\sigma_{p_2}^2)}{\sigma_c^2(n\sigma_{p_1}^2 + m\sigma_{p_2}^2) + \sigma_{p_1}^2 \sigma_{p_2}^2}\hat{\boldsymbol{\phi}}_j + \frac{\sigma_{p_2}^2(\sigma_c^2 - \tau^2)}{\sigma_c^2(n\sigma_{p_1}^2 + m\sigma_{p_2}^2) + \sigma_{p_1}^2 \sigma_{p_2}^2}\sum_{\substack{c_i \in \mathcal{C}_1 \\ i \neq j}}\boldsymbol{\psi}_i$$

$$+ \frac{\sigma_{p_1}^2(\sigma_c^2 - \tau^2)}{\sigma_c^2(n\sigma_{p_1}^2 + m\sigma_{p_2}^2) + \sigma_{p_1}^2 \sigma_{p_2}^2}\sum_{\substack{c_i \in \mathcal{C}_2 \\ i \neq j}}\boldsymbol{\psi}_i + \boldsymbol{p}_{\boldsymbol{\theta}_j^*}. \tag{47}$$

This is the Bayes optimal solution to the local Bayes objective optimization problem for client $c_j$ in (Local Bayes objective). Now, recall that in FEDHDP, the clients do not have access to individual client updates, but rather the global model. Therefore, the clients solve the FEDHDP local objective in (Local FedHDP objective). Given a value of $\lambda_j$ and the global estimate $\hat{\boldsymbol{\theta}}^*$, the minimizer $\hat{\boldsymbol{\theta}}_j(\lambda_j)$ of (Local FedHDP objective) is

$$\hat{\boldsymbol{\theta}}_j(\lambda_j) = \frac{1}{1 + \lambda_j}\Big( \hat{\boldsymbol{\phi}}_j + \lambda_j\hat{\boldsymbol{\theta}}^* \Big) \tag{48}$$

$$= \frac{1}{1 + \lambda_j}\Big( \frac{(N_1 + N_2 r) + \lambda_j i_j}{(N_1 + N_2 r)}\hat{\boldsymbol{\phi}}_j + \frac{\lambda_j}{(N_1 + N_2 r)}\sum_{\substack{c_i \in \mathcal{C}_1 \\ i \neq j}}\boldsymbol{\psi}_i + \frac{\lambda_j r}{(N_1 + N_2 r)}\sum_{\substack{c_i \in \mathcal{C}_2 \\ i \neq j}}\boldsymbol{\psi}_i \Big), \tag{49}$$

where $i_j = 1$ if $c_j \in \mathcal{C}_1$ or $i_j = r$ if $c_j \in \mathcal{C}_2$. Now, we are ready to state the Bayes optimality of the local FedHDP objective for optimal values $\lambda_j^*$ for all clients.

**Lemma 6** (Local estimates optimality). *The solution to the local FedHDP objective from the clients' point of view using $\lambda_j^*$ chosen below, under the assumption of global estimate optimality stated in Lemma 5, is Bayes optimal in the considered federated linear regression problem.*

$$\lambda_j^* = \begin{cases} \frac{N(1+\Upsilon^2)+N_1\Gamma_2^2+N_2\Gamma_1^2}{N\Upsilon^2(1+\Upsilon^2)+\Upsilon^2((N_2+1)\Gamma_1^2+N_1\Gamma_2^2)+\Gamma_1^2(1+\Gamma_2^2)}, & \text{if } c_j \in \mathcal{C}_1 \\ \frac{N(1+\Upsilon^2)+N_1\Gamma_2^2+N_2\Gamma_1^2}{N\Upsilon^2(1+\Upsilon^2)+\Upsilon^2(N_2\Gamma_1^2+(N_1+1)\Gamma_2^2)+\Gamma_2^2(1+\Gamma_1^2)}, & \text{if } c_j \in \mathcal{C}_2 \end{cases} \tag{50}$$

*where $\Upsilon^2 = \frac{\tau^2}{\alpha^2}$, $\Gamma_1^2 = \frac{N_1\gamma_1^2}{\alpha^2}$, and $\Gamma_2^2 = \frac{N_2\gamma_2^2}{\alpha^2}$.*

*Proof.* To prove this lemma, as shown in (Li et al., 2021), we only need to find the optimal values of $\lambda_j^*$ that minimize the following

$$\lambda_j^* = \arg\min_\lambda \mathbb{E}\big(\|\boldsymbol{\theta}_j^* - \hat{\boldsymbol{\theta}}_j(\lambda)\|_2^2 \big| \boldsymbol{\phi}^{\backslash j}, \hat{\boldsymbol{\phi}}_j\big) \tag{51}$$

for private and non-private clients. To compute the values of $\lambda_j^*$, we plug in the values of $\boldsymbol{\theta}_j^*$ from (47) and $\boldsymbol{\theta}_j(\lambda)$ in (49), which gives us the following

$$\lambda_1 = \frac{(N_1 + N_2 r)\big(\sigma_c^2(n\sigma_{p_1}^2 + m\sigma_{p_2}^2) - \tau^2(n\sigma_{p_1}^2 + m\sigma_{p_2}^2)\big)}{(N_1 + N_2 r)\big(\tau^2(n\sigma_{p_1}^2 + m\sigma_{p_2}^2) + \sigma_{p_1}^2\sigma_{p_2}^2\big) - i_j\big(\sigma_c^2(n\sigma_c^2 + m\sigma_p^2) + \sigma_{p_1}^2\sigma_{p_2}^2\big)}, \tag{52}$$

$$\lambda_2 = \frac{(N_1 + N_2 r)(\sigma_c^2 - \tau^2)\sigma_{p_2}^2}{\sigma_c^2(n\sigma_{p_1}^2 + m\sigma_{p_2}^2) + \sigma_{p_1}^2\sigma_{p_2}^2 - (N_1 + N_2 r)(\sigma_c^2 - \tau^2)\sigma_{p_2}^2}, \tag{53}$$

$$\lambda_3 = \frac{(N_1 + N_2 r)(\sigma_c^2 - \tau^2)\sigma_{p_2}^2}{\sigma_c^2(n\sigma_{p_1}^2 + m\sigma_{p_2}^2) + \sigma_{p_1}^2\sigma_{p_2}^2 - (N_1 + N_2 r)(\sigma_c^2 - \tau^2)\sigma_{p_2}^2}, \tag{54}$$

$$\text{and } \lambda_j^* = \frac{1}{3}(\lambda_1 + \lambda_2 + \lambda_3) \tag{55}$$

where $r = \frac{\sigma_{p_1}^2}{\sigma_{p_2}^2}$. For client $c_j \in \mathcal{C}_1$, we have $i_j = 1, n = N_2$ and $m = N_1 - 1$. Setting $\Upsilon^2 = \frac{\tau^2}{\alpha^2}$, $\Gamma_1^2 = \frac{N_1\gamma_1^2}{\alpha^2}$, and $\Gamma_2^2 = \frac{N_2\gamma_2^2}{\alpha^2}$ and substituting in (55) gives the desired result in (50). For client $c_j \in \mathcal{C}_2$, we have $i_j = r, n = N_2 - 1$ and $m = N_1$. Setting $\Upsilon^2 = \frac{\tau^2}{\alpha^2}$, $\Gamma_1^2 = \frac{N_1\gamma_1^2}{\alpha^2}$, and $\Gamma_2^2 = \frac{N_2\gamma_2^2}{\alpha^2}$ and substituting in (55) gives the desired results in (50). As a result, the resulting $\hat{\boldsymbol{\theta}}_j(\lambda_j^*)$ is Bayes optimal. $\qquad\square$

Next, we provide a few examples of corner cases for both $\lambda_p^*$ and $\lambda_{np}^*$ for the considered linear regression setup:

- $r \to 1$, i.e., noise added for privacy is similar for both sets of clients $\Gamma_1^2 \to \Gamma_2^2$, $\lambda_1^* \to \frac{N(1+\Upsilon^2) + N\Gamma_2^2}{N\Upsilon^2(1+\Upsilon^2) + (N+1)\Upsilon^2\Gamma_2^2 + (1+\Gamma_2^2)\Gamma_2^2}$ and $\lambda_2^* \to \frac{N(1+\Upsilon^2) + N\Gamma_2^2}{N\Upsilon^2(1+\Upsilon^2) + (N+1)\Upsilon^2\Gamma_2^2 + (1+\Gamma_2^2)\Gamma_2^2}$. If $\Gamma_1^2 = \Gamma_2^2 = 0$, then we have $\lambda_1^* = \lambda_2^* \to \frac{1}{\Upsilon^2}$ as in DITTO with FEDAVG and no malicious clients.

- $N_2 \to N$, i.e., all clients have the same privacy level, as in DP-FEDAVG, $\lambda_2^* \to \frac{N}{\Upsilon^2 N + \Gamma_2^2}$.

- $\alpha^2 \to 0$, $\lambda_2^* \to 0$ and $\lambda_1^* \to 0$. The optimal estimator for all clients approaches the local estimator, i.e., $\hat{\boldsymbol{\theta}}_j(\lambda_j^*) \to \hat{\boldsymbol{\phi}}_j$.

- $\tau^2 \to 0$, i.e., all clients have IID samples, $\lambda_1^* \to \frac{N + N_2\Gamma_1^2 + N_1\Gamma_2^2}{\Gamma_1^2(1+\Gamma_2^2)}$ and $\lambda_2^* \to \frac{N + N_2\Gamma_1^2 + N_1\Gamma_2^2}{\Gamma_2^2(1+\Gamma_1^2)}$.

## A.4 Optimality of FedHDP

Next, we show the convergence of the FEDHDP algorithm to the FEDHDP global and local objectives for the linear regression problem described above as follows

**Lemma 7** (FedHDP convergence). *FedHDP, with learning rate $\eta = 1$ and $\eta_p = \frac{1}{1+\lambda_j}$ converges to the global FedHDP objective and the local FedHDP objective.*

*Proof.* In the considered setup, we denote $\hat{\boldsymbol{\phi}}_j = \frac{1}{n_s}\sum_{i=1}^{n_s} \boldsymbol{x}_{j,i}$ at client $c_j$. The client updates the global estimation $\boldsymbol{\theta}$ by minimizing the loss function in (23). The global estimation update at the client follows

$$\boldsymbol{\theta} \leftarrow \boldsymbol{\theta} - \eta(\boldsymbol{\theta} - \hat{\boldsymbol{\phi}}_j). \tag{56}$$

Updating the estimation once with $\eta = 1$ results in the global estimation update being $\hat{\boldsymbol{\phi}}_j$, adding the noise results in the same $\boldsymbol{\psi}_j$, and hence the global estimate in the next iteration is unchanged. As for the local FedHDP estimation, when the client receives the global estimate $\boldsymbol{\theta}$ after the first round, the client updates its estimate $\boldsymbol{\theta}_j$ as

$$\boldsymbol{\theta}_j \leftarrow \boldsymbol{\theta}_j - \eta_p\big((\boldsymbol{\theta}_j - \hat{\boldsymbol{\phi}}_j) + \lambda_j(\boldsymbol{\theta}_j - \boldsymbol{\theta})\big). \tag{57}$$

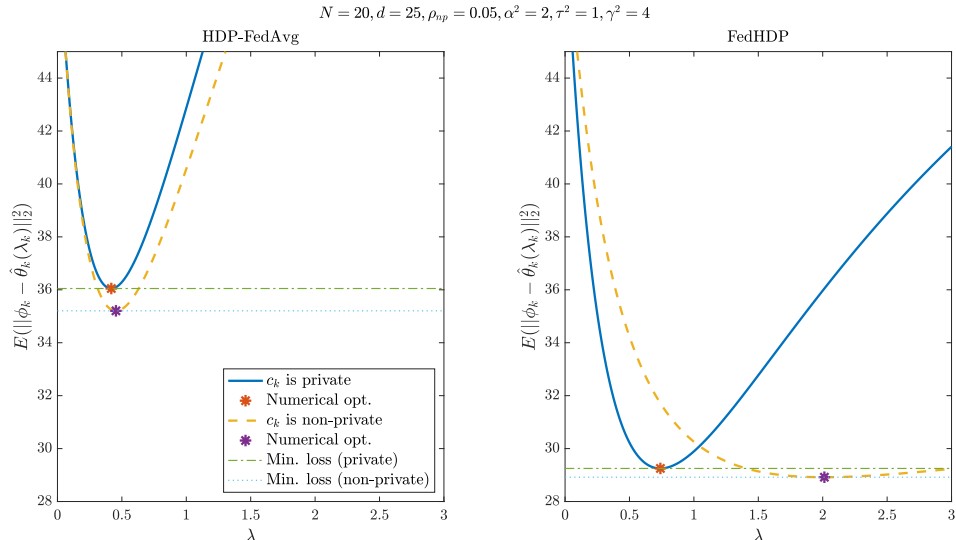

Figure 3: The effect of opting out on the personalized local model estimate for a linear regression problem as a function of $\lambda$ when employing (left) HDP-FEDAVG and (right) FEDHDP.

Updating the estimate once with $\eta_p = \frac{1}{1+\lambda_j}$ gives $\boldsymbol{\theta}_j = \frac{1}{1+\lambda_j}(\hat{\boldsymbol{\phi}}_j + \lambda_j \boldsymbol{\theta})$, which is the solution to the local FedHDP objective in (48). Hence, FedHDP converges to the global and local FedHDP objectives. $\qquad \square$

Next, we state the optimality theorem of FEDHDP algorithm for the considered setup described above.

**Theorem 8** (FedHDP optimality)**.** *FedHDP from the server's point of view with ratio $r^*$ chosen below, is Bayes optimal (i.e., $\boldsymbol{\theta}$ converges to $\boldsymbol{\theta}^*$) in the considered federated linear regression problem.*

$$r^* = \frac{\sigma_c^2 + N_1 \gamma_1^2}{\sigma_c^2 + N_2 \gamma_2^2}. \tag{58}$$

*Furthermore, FedHDP from the clients point of view, with $\lambda_j^*$ chosen below, is Bayes optimal (i.e., $\boldsymbol{\theta}_j$ converges to $\boldsymbol{\theta}_j^*$ for each client $j \in [N]$) in the considered federated linear regression problem.*

$$\lambda_j^* = \begin{cases} \frac{N(1+\Upsilon^2) + N_1 \Gamma_2^2 + N_2 \Gamma_1^2}{N\Upsilon^2(1+\Upsilon^2) + \Upsilon^2((N_2+1)\Gamma_1^2 + N_1\Gamma_2^2) + \Gamma_1^2(1+\Gamma_2^2)}, & \text{if } c_j \in \mathcal{C}_1 \\ \frac{N(1+\Upsilon^2) + N_1 \Gamma_2^2 + N_2 \Gamma_1^2}{N\Upsilon^2(1+\Upsilon^2) + \Upsilon^2(N_2\Gamma_1^2 + (N_1+1)\Gamma_2^2) + \Gamma_2^2(1+\Gamma_1^2)}, & \text{if } c_j \in \mathcal{C}_2 \end{cases} \tag{59}$$

*Proof.* Follows by observing Lemma 7, which states that the algorithm converges to the global and local FedHDP objectives, then by Lemma 5 and Lemma 6, which state that the solution to the FedHDP objective is the Bayes optimal solution for both global and local objectives. $\qquad \square$

### A.5 Privacy-Utility Tradeoff

Let us consider the special case of opt-out of privacy in this simplified linear regression setup, i.e., $\gamma_1^2 = 0$ and $\gamma_2^2 = \gamma^2$. As discussed in the paper for federated point estimation, we would like to observe the effect of opting out of privacy on the client's personalized local model, compared to the one where the client remains private. We show an experiment comparing FEDHDP using $r^*$ against HDP-FEDAVG for two scenarios. The first is when the client chooses to opt out of privacy, and the second is when the client chooses to remain private. See Figure 3 for the results of such experiment. We can see that FEDHDP outperforms the one with HDP-FEDAVG, and the cost of remaining private is evident in terms of higher loss at the client.

### A.6 Extension Beyond Two Privacy Levels

The setup for federated learning with two privacy levels was presented to show the steps and the explicit expressions for the values of the ratio and the regularization hyperparameters. Next, we show briefly that

the same derivation can be extended to find the explicit expressions for the hyperparameters in a general case of federated linear regression with clients choosing one of $l$ privacy levels. To start, assume that we have $l$ privacy levels where clients can be split into $l$ subsets denoted by $\mathcal{C}_i$ for $i = 1, 2, ..., l$, each has $N_i > 1$ clients, respectively, while other notations are still the same. Notice that this setup contains the most general case where $l = |\mathcal{C}|$ and each client has their own privacy level.

Similar to the setup with two privacy levels, each client sends their update to the server after adding the appropriate amount of Gaussian noise, i.e., $\mathcal{N}(\mathbf{0}, N_i \gamma_i^2 I_d)$ for client $c_j \in \mathcal{C}_i$, for privacy. Let us denote the updates sent to the server by $\{\psi : i \in [N]\}$ which are estimates of $\phi$ with zero-mean Gaussian noise with corresponding covariance matrices $\{\Sigma_i : i \in [N]\}$, where $\Sigma_j$ for client $c_j \in \mathcal{C}_i$ is expressed as

$$\Sigma_j = (\alpha^2 + \tau^2 + N_i \gamma_i^2) I_d = \sigma_j^2 I_d = \sigma_{p_i}^2 I_d. \tag{60}$$

Then, the server computes its estimate $\boldsymbol{\theta}$ of $\boldsymbol{\phi}$ as

$$\boldsymbol{\theta} = \Sigma_{\boldsymbol{\theta}} \sum_{i \in [N]} \Sigma_i^{-1} \psi_i + \boldsymbol{p_\theta}, \tag{61}$$

where $\boldsymbol{p_\theta} \sim \mathcal{N}(\mathbf{0}, \Sigma_\theta)$ and

$$\Sigma_{\boldsymbol{\theta}} = \Big( \sum_{i \in [N]} \Sigma_i^{-1} \Big)^{-1} = \Big( \frac{\prod_{i \in [l]} \sigma_{p_i}^2}{\sum_{i \in [l]} N_i \prod_{k \in [l] \setminus i} \sigma_{p_k}^2} \Big) I_d. \tag{62}$$

In FEDHDP, the server applies a weighted averaging to the clients' updates $\psi_i$'s of this form

$$\boldsymbol{\theta} = \sum_{i \in [N]} w_i \psi_i. \tag{63}$$

To achieve the optimal covariance of the estimation at the server, the resulting weights used for client $c_j$ in $\mathcal{C}_i$ at the server as follows

$$w_j = \frac{r_i}{\sum_{k=[l]} N_k r_k}. \tag{64}$$

In this case, we have $l$ ratio hyperparameters $r_i$'s to tune. Similar to the approach followed for the 2-level privacy heterogeneity, we can find the optimal values of the ratio hyperparameters that achieve the optimal covariance of the estimation at the server. The optimal values of $r_i^*$ are

$$r_i^* = \frac{\sigma_{p_1}^2}{\sigma_{p_i}^2}. \tag{65}$$

Next, we compute the Bayes optimal local estimate $\boldsymbol{\theta}_j^*$ of $\boldsymbol{\phi}_j$ for the local objective at client $c_j$. We consider client $c_j$, which can be in any private set of clients $\mathcal{C}_i$, and compute their minimizer of (Local Bayes objective). In this case, the client has access to all other clients' private estimates $\{\psi_i : i \in [N] \setminus j\}$ and has their own local non-private estimate $\hat{\boldsymbol{\phi}}_j$. Similar to the approach before, we utilize Lemma 4 to find the optimal estimate $\boldsymbol{\theta}_j^*$. Given the updates by all other clients $\{\psi_i : i \in [N] \setminus j\}$, the client can compute the estimate $\phi^{\setminus j}$ of the value of $\phi$ as

$$\phi^{\setminus j} = \Sigma_{\boldsymbol{\phi}^{\setminus j}} \Big( \sum_{i=[N] \setminus j} \Sigma_i^{-1} \psi_i \Big) + \boldsymbol{p}_{\boldsymbol{\phi}^{\setminus j}}, \tag{66}$$

where $\boldsymbol{p}_{\boldsymbol{\phi}^{\setminus j}} \sim \mathcal{N}(\mathbf{0}, \Sigma_{\boldsymbol{\phi}^{\setminus j}})$ and

$$\Sigma_{\boldsymbol{\phi}^{\setminus j}} = \Big( \sum_{i=[N] \setminus j} \Sigma_i^{-1} \Big)^{-1}, \tag{67}$$

$$= \Big( M_1 \frac{1}{\sigma_{p_1}^2} I_d + M_2 \frac{1}{\sigma_{p_2}^2} I_d + ... + M_l \frac{1}{\sigma_{p_l}^2} I_d \Big)^{-1}, \tag{68}$$

$$= \frac{\prod_{i \in [l]} \sigma_{p_i}^2}{\sum_{i \in [l]} M_i \prod_{k \in [l] \setminus i} \sigma_{p_k}^2} I_d, \tag{69}$$

where

$$M_i = \begin{cases} N_i & \text{if } c_j \notin \mathcal{C}_i \\ N_i - 1, & \text{if } c_j \in \mathcal{C}_i \end{cases}. \tag{70}$$

Therefore, we have the following

$$\boldsymbol{\phi}^{\setminus j} = \frac{1}{\sum_{i \in [l]} M_i \prod_{k \in [l] \setminus i} \sigma_{p_k}^2} \Big( \sum_{i=[l]} \prod_{k_1 \in [l] \setminus i} \sigma_{p_{k_1}}^2 \sum_{\substack{c_{k_2} \in \mathcal{C}_i \\ k_2 \neq j}} \boldsymbol{\psi}_i \Big) + \boldsymbol{p}_{\boldsymbol{\phi}^{\setminus j}}. \tag{71}$$

Then, the client uses $\Sigma_{\boldsymbol{\phi}^{\setminus j}}$ and $\hat{\boldsymbol{\phi}}_j$ to estimate $\boldsymbol{\theta}_j^*$ as

$$\boldsymbol{\theta}_j^* = \Sigma_{\boldsymbol{\theta}_j^*} \Big( \big( \big( \frac{\prod_{i \in [l]} \sigma_{p_i}^2}{\sum_{i \in [l]} M_i \prod_{k \in [l] \setminus i} \sigma_{p_k}^2} + \tau^2 \big) I_d \big)^{-1} \boldsymbol{\phi}^{\setminus j} + (\sigma_c^2 - \tau^2)^{-1} \hat{\boldsymbol{\phi}}_j \Big) + \boldsymbol{p}_{\boldsymbol{\theta}_j^*}, \tag{72}$$

where $\boldsymbol{p}_{\boldsymbol{\theta}_j^*} \sim \mathcal{N}(\mathbf{0}, \Sigma_{\boldsymbol{\theta}_j^*})$ and

$$\Sigma_{\boldsymbol{\theta}_j^*} = \Big( \Big( \frac{\prod_{i \in [l]} \sigma_{p_i}^2 + \tau^2 \sum_{i \in [l]} M_i \prod_{k \in [l] \setminus i} \sigma_{p_k}^2}{\sum_{i \in [l]} M_i \prod_{k \in [l] \setminus i} \sigma_{p_k}^2} \Big)^{-1} + (\sigma_c^2 - \tau^2)^{-1} \Big)^{-1} I_d. \tag{73}$$

We expand (72) as

$$\boldsymbol{\theta}_j^* = \Sigma_{\boldsymbol{\theta}_j^*} (\sigma_c^2 - \tau^2)^{-1} \hat{\boldsymbol{\phi}}_j + \Sigma_{\boldsymbol{\theta}_j^*} \Big( \frac{1}{\prod_{i \in [l]} \sigma_{p_i}^2 + \tau^2 \sum_{i \in [l]} M_i \prod_{k \in [l] \setminus i} \sigma_{p_k}^2} \sum_{i=[l]} \prod_{k_1 \in [l] \setminus i} \sigma_{p_{k_1}}^2 \sum_{\substack{c_{k_2} \in \mathcal{C}_i \\ k_2 \neq j}} \boldsymbol{\psi}_i \Big) + \boldsymbol{p}_{\boldsymbol{\theta}_j^*}. \tag{74}$$

This is the Bayes optimal solution to the local Bayes objective optimization problem for client $c_j$ in (Local Bayes objective). Next, we know in FEDHDP the clients do not have access to individual client updates, but rather the global model. As a result, the clients solve the FEDHDP local objective in (Local FedHDP objective). Given a value of $\lambda_j$ and the global estimate $\hat{\boldsymbol{\theta}}^*$, the minimizer $\hat{\boldsymbol{\theta}}_j(\lambda_j)$ of (Local FedHDP objective) is

$$\hat{\boldsymbol{\theta}}_j(\lambda_j) = \frac{1}{1 + \lambda_j} \Big( \hat{\boldsymbol{\phi}}_j + \lambda_j \hat{\boldsymbol{\theta}}^* \Big) \tag{75}$$

$$= \frac{1}{1 + \lambda_j} \Big( \frac{\sum_{i=[l]} N_i r_i + \lambda_j i_j}{\sum_{i=[l]} N_i r_i} \hat{\boldsymbol{\phi}}_j + \frac{1}{\sum_{i=[l]} N_i r_i} \sum_{i \in [l]} \lambda_j r_i \sum_{\substack{c_k \in \mathcal{C}_i \\ k \neq j}} \boldsymbol{\psi}_k \Big), \tag{76}$$

where $i_j = r_i$ for client $c_j \in \mathcal{C}_i$. Note that we have $l + 1$ terms in both (76) and (74), which we can use to compute the value of $\lambda_j^*$ as done in the previous section, and results similar to the ones in the prior parts of this appendix then follow from such findings. Note that computing the expressions of the optimal $\lambda_j^*$ in closed form for each one of the $l$ sets of private clients in the considered setup is involved; hence, our brief presentation of the sketch of the solution.

# B  Federated Point Estimation

In this section, we continue the discussion started in Section 3, and make use of the results stated in Appendix A. In the federated point estimation problem, $\boldsymbol{F}_j = [1, 1, ..., 1]^T$ of length $n_s$ at client $c_j$. The results in the previous section can be used for federated point estimation by using $d = 1$. In the remainder of this section, we assume the opt-out of privacy scenario where clients choose to be either private or non-private in the setup. First, we restate Lemma 1 and show its proof.

**Lemma 9** (Global estimate optimality (Lemma 1 restated)). *FEDHDP from the server's point of view, with ratio $r^*$ chosen below, is Bayes optimal (i.e., $\theta$ converges to $\theta^*$) in the considered federated point estimation problem given by $r^* = \frac{\sigma_c^2}{\sigma_c^2 + N_p \gamma^2}$. Furthermore, the resulting variance is:*

$$\sigma_{s,\text{opt}}^2 = \frac{1}{N} \left[ \frac{\sigma_c^2(\sigma_c^2 + N_p\gamma^2)}{\sigma_c^2 + \rho_{np}N_p\gamma^2} \right]. \tag{77}$$

*Proof.* Follows directly by setting $d = 1$, $\gamma_1^2 = 0$, $\gamma_2^2 = \gamma^2$, $N_1 = N_{np}$, and $N_2 = N_p$ in Theorem 8. $\square$

Next, we restate Theorem 3 and show its proof.

**Theorem 10** (Local estimate optimality (Theorem 3 restated)). *Assuming using FEDHDP with ratio $r^*$ in Lemma 1, and using the values $\lambda_{np}^*$ for non-private clients and $\lambda_p^*$ for private clients stated below, FEDHDP is Bayes optimal (i.e., $\theta_j$ converges to $\theta_j^*$ for each client $j \in [N]$)*

$$\lambda_{np}^* = \frac{1}{\Upsilon^2}, \tag{78}$$

$$\lambda_p^* = \frac{N + N\Upsilon^2 + (N - N_p)\Gamma^2}{N\Upsilon^2(\Upsilon^2 + 1) + (N - N_p + 1)\Upsilon^2\Gamma^2 + \Gamma^2}. \tag{79}$$

*where $\Upsilon^2 = \frac{\tau^2}{\alpha^2}$ and $\Gamma^2 = \frac{N_p\gamma^2}{\alpha^2}$.*

*Proof.* Follows directly by setting $d = 1$, $\Gamma_1^2 = 0$, $\Gamma_2^2 = \Gamma^2$, $N_1 = N_{np}$, and $N_2 = N_p$ in Theorem 8. $\square$

Finally, we show one additional simulation result for the federated point estimation problem. The resulting server noise $\sigma_s^2$ versus the fraction of non-private clients $\rho_{np}$ is plotted for two scenarios. The first is the baseline FEDAVG, and the second is the optimal FEDHDP. We can see in Figure 4 that FEDHDP provides better noise variance at the server compared to FEDAVG, and the gain can be significant for some values of $\rho_{np}$, even if small percentage of clients opt out.

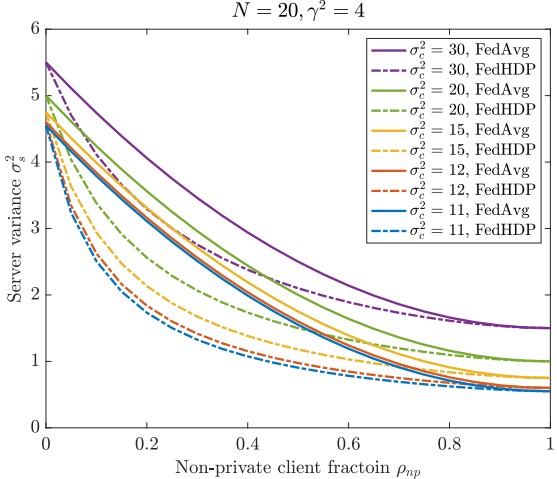

Figure 4: Server noise variance $\sigma_s^2$ vs non-private client fraction $\rho_{np}$ for the baseline FEDAVG aggregator and optimal FEDHDP aggregator.

## C   Experiments: Extended Experimental Results

In this section, we provide an extended version of the results of experiments conducted on the considered datasets. We describe the datasets along with the associated tasks in Tables 5, the models used in Table 6, and the hyperparameters used in Table 7.

Table 5: Experiments setup: Number of clients is $N$, approximate fraction of clients per round is $q$.

| Dataset | $N$ | $q$ | Task | Model |
|---|---|---|---|---|
| non-IID MNIST | 2,000 | 5% | 10-label classification | FC NN |
| FMNIST | 3,383 | 3% | 10-label classification | FC NN |
| FEMNIST | 3,400 | 3% | 62-label classification | CNN |

Table 6: Models used for experiments.

| non-IID MNIST, and FMNIST Datasets | | |
|---|---|---|
| **Layer** | **Size** | **Activation** |
| Input image | $28 \times 28$ | - |
| Flatten | 784 | - |
| Fully connected | 50 | ReLU |
| Fully connected | 10 | Softmax |
| FEMNIST Dataset | | |
| Input image | $28 \times 28$ | - |
| Convolutional (2D) | $28 \times 28 \times 16$ | ReLU |
| Max pooling (2D) | $14 \times 14 \times 16$ | - |
| Convolutional (2D) | $14 \times 14 \times 32$ | ReLU |
| Max pooling (2D) | $7 \times 7 \times 32$ | - |
| Dropout (25%) | - | - |
| Flatten | 1568 | - |
| Fully connected | 128 | ReLU |
| Dropout (50%) | - | - |
| Fully connected | 62 | Softmax |

Table 7: Hyperparameters used for each experiment.

| Hyperparameter | **non-IID MNIST** | **FMNIST** | **FEMNIST** |
|---|---|---|---|
| Batch size | 20 | | |
| Epochs | 25 | 50 | 25 |
| $\eta, \eta_p$ | 0.5 | 0.01 | 0.02 |
| $\eta, \eta_p$ decaying factor | 0.9 every 50 rounds | | N/A |
| $S^0$ | 0.5 | 0.5 | 2.0 |
| $\eta_b$ | 0.2 | | |
| $\kappa$ | 0.5 | | |
| Effective noise multiplier | 1.5 | 4.0 | 1.0 |

For each experiment, we presented the results of each dataset for the two baselines, i.e., Non-Private, DP-FedAvg, and HDP-FedAvg, as well as the proposed FedHDP algorithm along with the best parameters that produce the best results in the main body of the paper. In this appendix, we show the extended version of the experiments. For all experiments, training is stopped after 500 communication rounds for each experiment. The server's test dataset is the test MNIST dataset in the non-IID MNIST experiments, or the collection of the test datasets of all clients in the FMNIST and FEMNIST datasets. Note that the experiment of FedHDP with $r = 0$ denotes the case where the server only communicates with non-private clients during training and ignores all private clients. For the baseline algorithms, we note that the personalization scheme that is used on clients is Ditto. We vary the ratio hyperparameter $r$ as well as the regularization hyperparameters $\lambda_{\mathrm{p}}$ and $\lambda_{\mathrm{np}}$ at the clients and observe the results. In the following tables, we list the entirety of the results of all experiments conducted on each dataset for various values of the

hyperparameters. For readability, we highlight the rows that contain the best values of performance metrics in the proposed algorithm FedHDP.

Table 8: Experiment results on *non-IID MNIST*, $(\epsilon, \delta) = (3.6, 10^{-4})$. The variance of the performance metric across clients is between parenthesis.

| Setup | | Global model | | | | Personalized local models | | |
|---|---|---|---|---|---|---|---|---|
| **Algorithm** | **hyperparam.** | $Acc_g\%$ | $Acc_{\mathrm{g,p}}\%$ | $Acc_{\mathrm{g,np}}\%$ | $\triangle_{\mathrm{g}}\%$ | $Acc_{\mathrm{l,p}}\%$ | $Acc_{\mathrm{l,np}}\%$ | $\triangle_{\mathrm{l}}\%$ |
| | | | | $\lambda_{\mathrm{p}} = \lambda_{\mathrm{np}} = 0.005$ | | | | |
| Non-Private+Ditto | - | 93.8 | - | 93.75(0.13) | - | - | 99.98(0.001) | - |
| DP-FedAvg+Ditto | - | 88.75 | 88.64(0.39) | - | - | 99.97(0.002) | - | - |
| HDP-FedAvg+Ditto | - | 87.71 | 87.55(0.42) | 88.35(0.34) | 0.8 | 99.97(0.001) | 99.93(0.001) | −0.04 |
| FedHDP | $r=0$ | 90.7 | 90.64(0.68) | 91.72(0.5) | 1.08 | 90.64(0.68) | 99.94(0.001) | 9.2964 |
| FedHDP | $r=0.001$ | 91.74 | 91.65(0.39) | 92.61(0.27) | 0.94 | 99.94(0.001) | 99.95(0.001) | 0.01 |
| FedHDP | $r=0.01$ | 92.48 | 92.43(0.30) | 93.30(0.21) | 0.88 | 99.94(0.001) | 99.94(0.001) | 0 |
| FedHDP | $r=0.025$ | 92.36 | 92.28(0.27) | 92.96(0.19) | 0.68 | 99.95(0.001) | 99.91(0.001) | −0.04 |
| FedHDP | $r=0.1$ | 90.7 | 90.59(0.34) | 91.31(0.26) | 0.73 | 99.97(0.001) | 99.95(0.001) | −0.02 |
| | | | | $\lambda_{\mathrm{p}} = \lambda_{\mathrm{np}} = 0.05$ | | | | |
| Non-Private+Ditto | - | 93.81 | - | 93.76(0.13) | - | - | 99.93(0.001) | - |
| DP-FedAvg+Ditto | - | 87.98 | 87.97(0.39) | - | - | 99.84(0.002) | - | - |
| HDP-FedAvg+Ditto | - | 89.64 | 89.50(0.32) | 90.55(0.24) | 1.05 | 99.83(0.002) | 99.84(0.002) | 0.01 |
| FedHDP | $r=0$ | 91 | 91.12(0.48) | 92.08(0.41) | 0.96 | 91.12(0.48) | 99.76(0.002) | 8.65 |
| FedHDP | $r=0.001$ | 92.15 | 92.10(0.33) | 92.88(0.25) | 0.78 | 99.81(0.002) | 99.78(0.002) | −0.03 |
| FedHDP | $r=0.01$ | 92.45 | 92.39(0.33) | 93.26(0.25) | 0.87 | 99.81(0.002) | 99.78(0.003) | −0.03 |
| FedHDP | $r=0.025$ | 92.14 | 92.09(0.35) | 93.01(0.26) | 0.92 | 99.85(0.002) | 99.8(0.002) | −0.05 |
| FedHDP | $r=0.1$ | 90.7 | 90.82(0.29) | 91.55(0.21) | 0.73 | 99.87(0.002) | 99.80(0.003) | −0.06 |
| | | | | $\lambda_{\mathrm{p}} = \lambda_{\mathrm{np}} = 0.25$ | | | | |
| Non-Private+Ditto | - | 93.79 | - | 93.75(0.13) | - | - | 99.10(0.007) | - |
| DP-FedAvg+Ditto | - | 88.26 | 88.23(0.41) | - | - | 98.23(0.017) | - | - |
| HDP-FedAvg+Ditto | - | 88.27 | 88.09(0.49) | 89.08(0.4) | 0.99 | 98.14(0.017) | 98.13(0.017) | −0.01 |
| FedHDP | $r=0$ | 90.42 | 90.41(0.69) | 91.41(0.58) | 1.0 | 90.41(0.69) | 98.06(0.023) | 7.65 |
| FedHDP | $r=0.001$ | 92.18 | 92.12(0.34) | 92.85(0.26) | 0.73 | 98.47(0.015) | 98.08(0.024) | −0.39 |
| FedHDP | $r=0.01$ | 92.41 | 92.35(0.29) | 93.19(0.21) | 0.83 | 98.62(0.015) | 98.33(0.019) | −0.28 |
| FedHDP | $r=0.025$ | 92.5 | 92.42(0.28) | 93.19(0.19) | 0.77 | 98.71(0.011) | 98.41(0.017) | −0.3 |
| FedHDP | $r=0.1$ | 91.17 | 91.10(0.32) | 91.94(0.24) | 0.84 | 98.71(0.012) | 98.60(0.013) | −0.11 |

Table 9: Experiment results on *Skewed non-IID MNIST*, $(\epsilon, \delta) = (3.6, 10^{-4})$. The variance of the performance metric across clients is between parenthesis.

| Setup | | Global model | | | | Personalized local models | | |
|---|---|---|---|---|---|---|---|---|
| **Algorithm** | **hyperparam.** | $Acc_g\%$ | $Acc_{g,p}\%$ | $Acc_{g,np}\%$ | $\triangle_g\%$ | $Acc_{l,p}\%$ | $Acc_{l,np}\%$ | $\triangle_l\%$ |
| \multicolumn{9}{c}{$\lambda_p = \lambda_{np} = 0.005$} | | | | | | | | |
| Non-Private+Ditto | - | 93.67 | - | 93.62(0.15) | - | - | 99.98(0.001) | - |
| DP-FedAvg+Ditto | - | 88.93 | 88.87(0.35) | - | - | 99.98(0.001) | - | - |
| HDP-FedAvg+Ditto | - | 88.25 | 88.05(0.39) | 89.98(0.05) | 1.93 | 99.97(0.001) | 99.85(0.001) | −0.11 |
| FedHDP | $r=0$ | 10.27 | 7.1(6.5) | 100(0) | 92.9 | 7.1(6.5) | 100(0) | 92.9 |
| FedHDP | $r=0.025$ | 87.11 | 86.61(1.10) | 98.16(0.01) | 11.55 | 99.99(0.001) | 99.91(0.001) | −0.08 |
| FedHDP | $r=0.1$ | 90.36 | 89.96(0.37) | 97.45(0.01) | 7.49 | 99.97(0.001) | 99.76(0.003) | −0.21 |
| FedHDP | $r=0.5$ | 88.44 | 88.14(0.36) | 93.36(0.03) | 5.2 | 99.98(0.001) | 99.93(0.001) | −0.05 |
| FedHDP | $r=0.75$ | 89.14 | 88.92(0.37) | 92.43(0.06) | 3.5 | 99.97(0.001) | 99.93(0.001) | −0.04 |
| FedHDP | $r=0.9$ | 87.96 | 87.69(0.56) | 92.97(0.04) | 5.28 | 99.98(0.001) | 99.96(0.001) | −0.02 |
| \multicolumn{9}{c}{$\lambda_p = \lambda_{np} = 0.05$} | | | | | | | | |
| Non-Private+Ditto | - | 93.67 | - | 93.62(0.15) | - | - | 99.93(0.001) | - |
| DP-FedAvg+Ditto | - | 88.78 | 88.70(0.53) | - | - | 99.83(0.002) | - | - |
| HDP-FedAvg+Ditto | - | 88.33 | 88.11(0.46) | 91.67(0.04) | 3.56 | 99.87(0.001) | 99.61(0.001) | −0.26 |
| FedHDP | $r=0$ | 10.28 | 7.1(6.5) | 100(0) | 92.9 | 7.1(6.5) | 100(0) | 92.9 |
| FedHDP | $r=0.025$ | 87.92 | 87.45(0.99) | 98.1(0.01) | 10.65 | 99.95(0.001) | 99.75(0.003) | −0.2 |
| FedHDP | $r=0.1$ | 88.98 | 88.64(0.52) | 96.18(0.02) | 7.54 | 99.9(0.001) | 99.47(0.005) | −0.43 |
| FedHDP | $r=0.5$ | 88.22 | 87.9(0.38) | 93.43(0.03) | 5.33 | 99.85(0.002) | 99.42(0.008) | −0.42 |
| FedHDP | $r=0.75$ | 88.56 | 88.37(0.35) | 91.33(0.04) | 2.94 | 99.84(0.002) | 99.52(0.004) | −0.33 |
| FedHDP | $r=0.9$ | 89.19 | 88.97(0.4) | 92.24(0.03) | 3.27 | 99.88(0.001) | 99.58(0.005) | −0.3 |
| \multicolumn{9}{c}{$\lambda_p = \lambda_{np} = 0.25$} | | | | | | | | |
| Non-Private+Ditto | - | 93.67 | - | 93.62(0.15) | - | - | 99.09(0.007) | - |
| DP-FedAvg+Ditto | - | 87.78 | 87.71(0.53) | - | - | 98.15(0.02) | - | - |
| HDP-FedAvg+Ditto | - | 89.4 | 89.22(0.26) | 92.01(0.03) | 2.79 | 98.27(0.02) | 97.62(0.03) | −0.64 |
| FedHDP | $r=0$ | 10.27 | 7.1(6.5) | 100(0) | 92.9 | 7.1(6.5) | 100(0) | 92.9 |
| FedHDP | $r=0.025$ | 87.51 | 87.01(0.9) | 98.49(0.01) | 11.48 | 98.69(0.01) | 99.09(0.006) | −0.4 |
| FedHDP | $r=0.1$ | 89.05 | 88.66(0.54) | 96.8(0.02) | 8.14 | 98.69(0.012) | 98.55(0.008) | −0.13 |
| FedHDP | $r=0.5$ | 88.18 | 88.11(0.55) | 93.43(0.03) | 5.32 | 98.32(0.014) | 97.80(0.01) | −0.52 |
| FedHDP | $r=0.75$ | 87.96 | 87.8(0.33) | 92.58(0.03) | 4.78 | 98.25(0.017) | 97.5(0.02) | −0.75 |
| FedHDP | $r=0.9$ | 88.26 | 87.93(0.41) | 91.67(0.03) | 3.74 | 98.25(0.02) | 97.68(0.02) | −0.57 |

Table 10: Experiment results on *FMNIST*, $(\epsilon, \delta) = (0.6, 10^{-4})$. The variance of the performance metric across clients is between parenthesis.

| Setup | | Global model | | | | Personalized local models | | |
|---|---|---|---|---|---|---|---|---|
| **Algorithm** | **hyperparam.** | $Acc_g\%$ | $Acc_{\mathrm{g,p}}\%$ | $Acc_{\mathrm{g,np}}\%$ | $\triangle_{\mathrm{g}}\%$ | $Acc_{\mathrm{l,p}}\%$ | $Acc_{\mathrm{l,np}}\%$ | $\triangle_{\mathrm{l}}\%$ |
| colspan9 $\lambda_{\mathrm{p}} = \lambda_{\mathrm{np}} = 0.005$ |
| NON-PRIVATE+DITTO | - | 89.65 | - | 89.35(1.68) | - | - | 93.95(0.67) | - |
| DP-FEDAVG+DITTO | - | 71.76 | 71.42(2.79) | - | - | 91.01(0.94) | - | - |
| HDP-FEDAVG+DITTO | - | 75.87 | 75.77(2.84) | 74.41(2.8) | −1.36 | 90.45(1.02) | 92.32(0.8) | 1.87 |
| FEDHDP | $r=0$ | 81.78 | 80.73(2.45) | 89.35(1.5) | 8.62 | 80.73(2.4) | 95.80(0.39) | 15.06 |
| FEDHDP | $r=0.01$ | 85.38 | 84.61(2.05) | 89.3(1.26) | 4.69 | 93.26(0.74) | 95.94(0.41) | 2.67 |
| FEDHDP | $r=0.025$ | 85.7 | 84.93(1.97) | 89.58(1.29) | 4.65 | 93.04(0.76) | 95.22(0.54) | 2.18 |
| FEDHDP | $r=0.05$ | 85.21 | 84.68(1.99) | 86.22(1.76) | 1.54 | 92.87(0.74) | 95.40(0.51) | 2.53 |
| FEDHDP | $r=0.1$ | 81.76 | 81.45(2.45) | 81.96(1.84) | 0.51 | 92.47(0.78) | 94.83(0.52) | 2.36 |
| FEDHDP | $r=0.5$ | 78.19 | 78.02(2.59) | 76.48(3.02) | −1.53 | 91.08(0.94) | 92.59(0.83) | 1.51 |
| colspan9 $\lambda_{\mathrm{p}} = \lambda_{\mathrm{np}} = 0.05$ |
| NON-PRIVATE+DITTO | - | 89.65 | - | 89.35(1.68) | - | - | 94.53(0.59) | - |
| DP-FEDAVG+DITTO | - | 77.61 | 77.62(2.55) | - | - | 90.04(1.04) | - | - |
| HDP-FEDAVG+DITTO | - | 72.42 | 77.14(2.72) | 76.28(2.76) | −0.86 | 89.12(1.15) | 90.92(0.91) | 1.8 |
| FEDHDP | $r=0$ | 82.61 | 80.72(2.45) | 89.45(1.51) | 8.73 | 80.72(2.45) | 95.57(0.38) | 14.84 |
| FEDHDP | $r=0.01$ | 86.88 | 85.36(1.89) | 90.02(1.28) | 4.66 | 93.76(0.68) | 95.78(0.36) | 2.02 |
| FEDHDP | $r=0.025$ | 86.03 | 84.22(1.98) | 88.40(1.68) | 4.18 | 93.53(0.68) | 95.11(0.54) | 0.52 |
| FEDHDP | $r=0.05$ | 84.65 | 82.68(2.16) | 86.68(1.67) | 4.00 | 92.92(0.76) | 95.02(0.55) | 2.1 |
| FEDHDP | $r=0.1$ | 82.89 | 81.72(2.28) | 83.68(2.18) | 1.96 | 92.38(0.83) | 94.25(0.61) | 1.87 |
| FEDHDP | $r=0.5$ | 76.59 | 78.05(2.60) | 78.04(2.66) | −0.0041 | 89.63(1.10) | 91.67(0.84) | 2.04 |
| colspan9 $\lambda_{\mathrm{p}} = \lambda_{\mathrm{np}} = 0.25$ |
| NON-PRIVATE+DITTO | - | 89.66 | - | 89.36(1.69) | - | - | 94.32(0.64) | - |
| DP-FEDAVG+DITTO | - | 70.1 | 70.40(2.91) | - | - | 88.38(1.25) | - | - |
| HDP-FEDAVG+DITTO | - | 72.67 | 72.05(2.83) | 74.31(2.42) | 2.26 | 87.54(1.34) | 87.39(1.23) | −0.15 |
| FEDHDP | $r=0$ | 81.93 | 80.85(2.39) | 89.71(1.39) | 8.86 | 80.85(2.39) | 94.56(0.50) | 13.71 |
| FEDHDP | $r=0.01$ | 85.31 | 84.55(1.98) | 89.27(1.54) | 4.72 | 92.76(0.78) | 94.77(0.5) | 2.01 |
| FEDHDP | $r=0.025$ | 86.17 | 85.52(1.92) | 89.25(1.31) | 3.73 | 92.46(0.85) | 94.35(0.57) | 1.89 |
| FEDHDP | $r=0.05$ | 83.97 | 83.5(2.19) | 85.4(1.88) | 1.9 | 91.69(0.91) | 93.9(0.53) | 2.21 |
| FEDHDP | $r=0.1$ | 83.78 | 83.22(2.11) | 84.94(2.12) | 1.72 | 90.9(1.02) | 92.62(0.73) | 1.72 |
| FEDHDP | $r=0.5$ | 74.64 | 74.63(3.23) | 72.54(2.93) | −2.09 | 88.12(1.34) | 88.69(1.28) | 0.57 |

Table 11: Experiment results on *FEMNIST*, $(\epsilon, \delta) = (4.1, 10^{-4})$. The variance of the performance metric across clients is between parenthesis.

| Setup | | Global model | | | | Personalized local models | | |
|---|---|---|---|---|---|---|---|---|
| **Algorithm** | **hyperparam.** | $Acc_g\%$ | $Acc_{g,p}\%$ | $Acc_{g,np}\%$ | $\triangle_g\%$ | $Acc_{l,p}\%$ | $Acc_{l,np}\%$ | $\triangle_l\%$ |
| \multicolumn{9}{c}{$\lambda_p = \lambda_{np} = 0.005$} | | | | | | | | |
| Non-Private+Ditto | - | 81.56 | | 81.72(1.37) | - | | 73.86(1.5) | - |
| DP-FedAvg+Ditto | - | 75.39 | 76.1(1.73) | - | - | 71.3(1.47) | - | - |
| HDP-FedAvg+Ditto | - | 74.96 | 75.6(1.69) | 77.84(1.47) | 2.24 | 71.44(1.5) | 70.03(1.18) | −1.4 |
| FedHDP | $r=0$ | 72.77 | 75.34(2.6) | 85.7(1.14) | 10.36 | 75.34(2.6) | 72.2(1.22) | −3.14 |
| FedHDP | $r=0.001$ | 73.66 | 76.22(2.44) | 86.04(1.2) | 9.82 | 72.78(1.51) | 71.74(1.34) | −1.04 |
| FedHDP | $r=0.01$ | 74.75 | 77.16(2.26) | 86.4(1.07) | 9.24 | 73.24(1.52) | 71.49(1.29) | −1.76 |
| FedHDP | $r=0.025$ | 75.37 | 77.66(2.06) | 86.56(1) | 8.9 | 73.11(1.55) | 71.62(1.18) | −1.49 |
| FedHDP | $r=0.1$ | 76.47 | 77.99(1.68) | 84.36(1.31) | 6.37 | 72.3(1.5) | 69.93(1.15) | −2.37 |
| FedHDP | $r=0.5$ | 76.11 | 76.69(1.62) | 80.82(1.3) | 4.13 | 71.32(1.56) | 70.11(1.26) | −1.2 |
| \multicolumn{9}{c}{$\lambda_p = \lambda_{np} = 0.05$} | | | | | | | | |
| Non-Private+Ditto | - | 81.95 | | 82.09(1.38) | - | | 82.89(1.13) | - |
| DP-FedAvg+Ditto | - | 75.42 | 75.86(1.82) | - | - | 74.69(1.29) | - | - |
| HDP-FedAvg+Ditto | - | 75.12 | 75.87(1.65) | 78.59(1.58) | 2.72 | 74.67(1.34) | 75.95(1.12) | 1.28 |
| FedHDP | $r=0$ | 72.65 | 75.9(2.5) | 86.19(1.27) | 10.29 | 80.59(1.13) | 81.97(0.88) | 1.38 |
| FedHDP | $r=0.001$ | 73.31 | 75.9(2.5) | 86.19(1.27) | 10.29 | 80.59(1.13) | 81.97(0.88) | 1.38 |
| FedHDP | $r=0.01$ | 74.68 | 77.16(2.27) | 86.25(1.05) | 9.09 | 80.74(1.06) | 82.13(0.98) | 1.38 |
| FedHDP | $r=0.025$ | 75.22 | 77.43(2.09) | 85.95(1.12) | 8.52 | 80(1.16) | 80.99(0.92) | 1.01 |
| FedHDP | $r=0.1$ | 76.52 | 77.91(1.67) | 83.9(1.27) | 5.99 | 77.9(1.22) | 79.15(0.99) | 1.25 |
| FedHDP | $r=0.5$ | 76.15 | 76.55(1.68) | 80.04(1.62) | 3.49 | 75.43(1.25) | 77.13(1.17) | 1.7 |
| \multicolumn{9}{c}{$\lambda_p = \lambda_{np} = 0.25$} | | | | | | | | |
| Non-Private+Ditto | - | 81.66 | | 81.79(1.38) | - | | 84.46(0.89) | - |
| DP-FedAvg+Ditto | - | 75.99 | 76.56(1.6) | - | - | 73.06(1.46) | - | - |
| HDP-FedAvg+Ditto | - | 75.31 | 75.67(1.71) | 78.88(1.59) | 3.21 | 72.58(1.45) | 74.98(1.43) | 2.4 |
| FedHDP | $r=0$ | 72.89 | 75.5(2.56) | 86.09(1.28) | 10.6 | 75.5(2.56) | 84.77(0.8) | 9.28 |
| FedHDP | $r=0.001$ | 73.41 | 76.01(2.51) | 85.99(1.13) | 9.97 | 80.98(1.06) | 84.71(0.83) | 3.73 |
| FedHDP | $r=0.01$ | 74.86 | 77.31(2.18) | 86.73(0.98) | 9.42 | 81.19(1.02) | 84.68(0.78) | 3.49 |
| FedHDP | $r=0.025$ | 75.41 | 77.68(2.1) | 86.23(1.03) | 8.55 | 80.01(1.1) | 83.2(0.8) | 3.19 |
| FedHDP | $r=0.1$ | 76.62 | 77.82(1.68) | 83.35(1.27) | 5.52 | 76.99(1.24) | 78.96(1.04) | 1.97 |
| FedHDP | $r=0.5$ | 75.89 | 76.71(1.65) | 80.01(1.4) | 3.3 | 73.48(1.37) | 75.49(1.57) | 2.01 |

