# OpenReview forum: "Federated Learning with Heterogeneous Differential Privacy"
_TMLR — Rejected by TMLR_

### Review · Reviewer_fZ6L · 2022-08-09

**Summary Of Contributions:**

In this paper, authors propose a differentially private (DP) federated learning (FL) algorithm for a setting where the clients have different privacy requirements. Since these varying privacy levels will lead to different levels of perturbation, authors propose a new weighted averaging method for aggregating the client side updates in to the global model. For setting where there are clients opting for two distinct privacy levels, authors show that the solution is Bayes optimal. Authors also consider the personalized client side models, and show theoretically (Thm 3) how the client side should combine the updates when there are clients opting in and out of DP. The theoretical analysis of this paper extends to point estimation and linear regression. In experiments authors demonstrate the effectiveness of their method in a classification task, and demonstrate that when the weights in the proposed averaging method FedHDP are tuned correctly, the proposed method out-performs the previous approach FedAvg which uses uniform averaging.

**Broader Impact Concerns:**

I don't think there are concerns that would require a Broader Impact Statement in this paper.

**Requested Changes:**

**Most critical things: (I want these to be clarified):**
* I'm having somewhat hard time following the math. I do understand, that the $\phi$ and $\phi_j$ you have used in equations (7) and (8) are functions of $\psi$'s, but it would be really good if you would be more explicit about this. Now it is completely readers burden to fill in the gaps in the details.
* The $\lambda_j$ in Equation (Local FedHDP objective) is not explained at all, and is not present in the "global" version above (eq. (9)). Please explain what $\lambda$ is. (I guess it is some type of regularization parameter but why is it only present in clients end and not in global model?)
* Before eq. (9) you say "... case of regression ... will be presented in the next section". Which Section are you referring to? In 3.2 there is no word about linear regression. Also Section 4 does not go any deeper on the details of this model.
* Setup in A.1:
    * You assume that the $F_j^T F_j$ is diagonal, which implies that the empirical covariance of the covariates is diagonal. Do you relax this assumption somewhere? If not, then you need to be more transparent in the main text about this very strong assumption. (IMPORTANT)
    * I don't quite understand the notation in (21). To me, the sum returns a scalar while $\mathbf{\phi}$ is a vector.
* Actually, I wonder is there something missing from equation (21). From (20) we can see, that even if we have the diagonal $F_j^T F_j$, the optimal solution should still depend on $F_j$. However, the $F_j$ seems to be completely missing from (21). I think you need to assume something more about $F_j$ to obtain this. (IMPORTANT)
* In proof of Lemma 5, you state "Lemma 4 states that the optimal aggregator at the server is the weighted average of the client updates.". This is not immediately clear to me since Lemma 4 does not speak anything about optimality.


**Recommendations that would strengthen the work:**
*  This paper seems like a highly relevant reference: Junxu Liu, Jian Lou, Li Xiong, Jinfei Liu, and Xiaofeng Meng. 2021. Projected federated averaging with heterogeneous differential privacy. Proc. VLDB Endow. 15, 4 (December 2021), 828–840. https://doi.org/10.14778/3503585.3503592
* In 3.2 you state: "Moreover, an even better and more realistic solution". I don't quite understand how this differs from the setting discussed just before if you assume that clients make "informed decisions". If the clients make informed decisions, then I would imagine these settings would end up the same.
* Use of "differentially-private datasets", "differentially-private local datasets": I think you mean privatized statistics/updates from DP clients? I think it might be good to be specific about to not be confused with DP data release techniques (the synthetic data techniques such as DP-GAN).
* Setup in A.1:
    * Should there be somekind of sub/supscript for $m_s$ and $c_j$ to relate them to the $C_1$ and $C_2$ you introduce earlier? Or does it matter for the setting?
    * For eq. (18) to make sense, I guess you need to assume that clients come from the same population? Otherwise I don't see how the $p_j$ could have zero mean.
    * In (19), you could also bold the argument for $f_j$
    * The $n_s$ e.g. in equation (21): should it be $m_s$ which you have introduced as the number of client's samples in the beginning of A.1?
* Please, connect (23) and (27). For both you state that these are the global objectives that the server aims to estimate, but you give two different expressions! I guess the (27) is the more useful one which actually relates the $\psi$'s from the clients to the $\phi$ you had in (23).
* About the points raised in Remark of Section 3.4
    * For 2nd: Why would we want to improve performance of malicious clients?
    * For 3rd: To me the "unaware of the status ... i.e. ... malicious"  and the privacy level requirements seem completely disconnected.

**Some typos:**
- "In the our discussion so far" >> "In our discussion so far"
- "spacial case"
- "First, The server-side"

**Strengths And Weaknesses:**

**Strengths:**
* ++ Strong empirical performance
* \+ Theoretical analysis for the linear regression example

**Weaknesses:**
* --- Presentation. In the current form it is really difficult (at least for me) to follow any of the technical details of Sections 3.1-3.5.
* --  Theoretical analysis for linear regression makes some strong assumptions (diagonal covariate covariance)
* --  Theoretical analysis seems to only extend to really simple settings (essentially only considers two privacy levels)
* \- Some unclear bits in the proofs (I will list in the requested changes)

---

> ### Author Response · Authors · 2022-08-26
> **Response to Reviewer fZ6L: Part 3**
>
> > * Setup in A.1:
> > * * Should there be some kind of sub/supscript for $m_s$ and $c_j$ to relate them to the $\mathcal{C}_1$ and $\mathcal{C}_2$ you introduce earlier? Or does it matter for the setting?
>
> We would like to point out that in this case, we only have a single round of training, so there is no need to index $c_j$ and $m_s$. Also, client $c_j$ can be in either $\mathcal{C}_1$ or $\mathcal{C}_2$.
>
> > * * For eq. (18) to make sense, I guess you need to assume that clients come from the same population? Otherwise I don't see how the pj could have zero mean.
>
> There are two hierarchies here: First hierarchy is a latent $\phi$, and the clients each have their own ground truth parameter $\phi_j$ which are correlated with each other through $\phi$. Then, the client $c_j$ obtains noisy observations from $\phi_j$ and that becomes their dataset; and the goal of client $c_j$ is to estimate $\phi_j$. The variance of $p_j$, i.e., $\tau^2$, here controls the degree of heterogeneity. If $\tau^2 = 0,$ we get $\phi_j = \phi$ for all clients and hence they all share the same goal. On the other hand, as $\tau^2$ grows, the tasks of the clients become unrelated to each other. Please note that this is just one way of modeling the data heterogeneity in data across clients, which happens to simplify the analysis to derive insights about heterogeneous data. We have added more discussion in the paper (Section 3.1) to clarify the setup.
>
> > * * In (19), you could also bold the argument for fj
>
> Thank you. Fixed.
>
> > * * The $n_s$ e.g. in equation (21): should it be $m_s$ which you have introduced as the number of client's samples in the beginning of A.1?
>
> It is correct in the text, because the problem is now translated from linear regression using $m_s$ samples to estimating a vector from $n_s$ vectors due to the $F^T F$ matrix being diagonal. In the case of point estimation, $n_s = m_s$. Please also note that $n_s$ is defined in the line before (23) in the revised version.
>
> > * Please, connect (23) and (27). For both you state that these are the global objectives that the server aims to estimate, but you give two different expressions! I guess the (27) is the more useful one which actually relates the ψ's from the clients to the ϕ you had in (23).
>
> To clarify, the two objectives are the Bayes objective and the FedHDP objective, respectively. So, from the server’s point of view, the aim is to utilize the updates from clients to estimate the value of $\boldsymbol{\phi}$, as in (23) (now (25) in the revised version). However, the FedHDP objective is to perform a weighted average to these updates such that the error is minimized, as in (27) (now (29) in the revised version). Our goal is to show that the solution to the FedHDP objective in the latter equation converges to the Bayes optimal solution in the former equation. We have addressed this issue by adding a clarification of this part in the revised version.
>
> > * About the points raised in Remark of Section 3.4
> > * * For 2nd: Why would we want to improve performance of malicious clients?
> > * * For 3rd: To me the "unaware of the status ... i.e. ... malicious" and the privacy level requirements seem completely disconnected.
>
> In the remark, we are highlighting the differences between the proposed FedHDP setup and the Ditto setup to differentiate the two setups in a clear manner. The Ditto setup is concerned with malicious vs benign clients in terms of robustness, while we are concerned with private vs non-private clients. One of the differences is that Ditto aims to improve the performance of benign clients, while in our setup we aim to improve the performance of non-private and private clients. Another difference is that in the Ditto setup, there are no additional information about the status of clients, i.e., malicious or benign, while in our setup the status of private and non-private clients is known to the server. In other words, the two setups are fundamentally different, and that is what we are discussing in the remark. We have revised some parts of the remark to clarify some possible confusing sentences.
>
> > * Some typos:\
> > "In the our discussion so far" >> "In our discussion so far"
> > "spacial case"
> > "First, The server-side"
>
> We have proofread the paper to correct these and other typos.

---

> ### Author Response · Authors · 2022-08-26
> **Response to Reviewer fZ6L: Part 2**
>
> > * Setup in A.1:
> > * * You assume that the $F_j^T F_j$ is diagonal, which implies that the empirical covariance of the covariates is diagonal. Do you relax this assumption somewhere? If not, then you need to be more transparent in the main text about this very strong assumption. (IMPORTANT)
>
> Thank you for highlighting this assumption that was not clearly highlighted. Although we mention it in the setup of the federated linear regression, we agree that it should be highlighted more clearly. To address this concern, we have clarified it in the revised version (Section 3.2) when discussing the extensions of the federated point estimation to more complex models, as well as upfront in the setup for the federated linear regression (Appendix A). We have also stated this as one of the limitations at the end of the paper.
>
>
> > * * I don't quite understand the notation in (21). To me, the sum returns a scalar while $\phi$ is a vector. Actually, I wonder is there something missing from equation (21). From (20) we can see, that even if we have the diagonal $F_j^TF_j$, the optimal solution should still depend on $F_j$. However, the $F_j$ seems to be completely missing from (21). I think you need to assume something more about Fj to obtain this. (IMPORTANT)
>
> The sum in (21) (now (23) in the revised version) in fact returns a vector, because as we observe the diagonal covariate covariance property, i.e., $F^TF=n_sI_d$, we can split the observations at the client $c_j$ into $n_s$ vectors, which we denote by $\boldsymbol{x}_{j,i}$’s as mentioned after now equation (23). As for the comment on $F_j$ missing from (21), we would like to clarify it is implicitly involved in the solution to the objective function. We can see that
> $\hat{\boldsymbol{\phi}}_j = \big( F_j^T F_j \big)^{-1} F_j^T \boldsymbol{x}_j$
> which we translate into the sum in equation (23) in the revised paper, which we could not reproduce in this response due to an OpenReview error. This translation holds because of the diagonal covariate covariance property. We revised the paper to clarify these intermediate steps.
>
> > * In proof of Lemma 5, you state "Lemma 4 states that the optimal aggregator at the server is the weighted average of the client updates.". This is not immediately clear to me since Lemma 4 does not speak anything about optimality.
>
> When we have $N$ unbiased noisy observations from a quantity, where those observations have different variances, Lemma 4 gives the optimal weighting of each observation so that the weighted aggregation has the minimal variance. Since this is Bayes optimal, there is no nonlinear aggregation of those observations that can possibly beat the simple weighted aggregation in terms of the resulting variance.
>
> > **Recommendations that would strengthen the work:**
> > * This paper seems like a highly relevant reference: Junxu Liu, Jian Lou, Li Xiong, Jinfei Liu, and Xiaofeng Meng. 2021. Projected federated averaging with heterogeneous differential privacy. Proc. VLDB Endow. 15, 4 (December 2021), 828–840. <https://doi.org/10.14778/3503585.3503592>
>
> Thank you for pointing this out. We have added this reference to the related work part in the revised version.
>
> > * In 3.2 you state: "Moreover, an even better and more realistic solution". I don't quite understand how this differs from the setting discussed just before if you assume that clients make "informed decisions". If the clients make informed decisions, then I would imagine these settings would end up the same.
>
> Thank you for pointing this out. We have revised this part of the paper to have a more coherent discussion around the setup, which can be found in Section 3.1.1. Please let us know if the revised version is still unclear.
>
>
> > * Use of "differentially-private datasets", "differentially-private local datasets": I think you mean privatized statistics/updates from DP clients? I think it might be good to be specific about to not be confused with DP data release techniques (the synthetic data techniques such as DP-GAN).
>
> Thank you for pointing this out to us. We agree with you and have corrected this in the revised version.

---

> ### Author Response · Authors · 2022-08-26
> **Response to Reviewer fZ6L: Part 1**
>
> We appreciate the reviewer’s comments on the paper, and highlighting the strengths of the theoretical analysis of the setup as well as the empirical experiments. Next, we include our response to the individual comments.
>
>
> > **Weaknesses:**
> > * Presentation. In the current form it is really difficult (at least for me) to follow any of the technical details of Sections 3.1-3.5.
>
> We appreciate the concern on the difficulty of following Section 3. We have made an effort to streamline the ideas in that section and simplify many expressions and notations. We have also added a table of notation in Table 1 to make it easier to refer to different quantities when reading the paper. We hope the revised version has better readability and is easier to follow. Please let us know if it needs further clarification.
>
>
>
> > * Theoretical analysis for linear regression makes some strong assumptions (diagonal covariate covariance)
>
> We have straightforwardly presented the assumptions made in the linear regression setup to remove any ambiguity about it, and we have clarified some of the confusing points in the analysis. We will discuss our modifications in the requested changes part of the response.
>
>
> > * Theoretical analysis seems to only extend to really simple settings (essentially only considers two privacy levels)
>
> We have extended the theoretical analysis to more than 2 privacy levels in the appendix of the revised version (appendix A.6). We have discussed the optimal server-side averaging process and provided the values of the optimal weighted aggregation hyperparameters. As for the regularization hyperparameters, we have provided a sketch of the solution as it is quite involved to compute such expressions for each set of clients in closed form with little insight expected to be observed from such a solution.
>
>
> > * Some unclear bits in the proofs (I will list in the requested changes)
>
> We will address this comment in the requested changes part.
>
>
> > **Requested Changes:**
>
> > **Most critical things: (I want these to be clarified):**
> > * I'm having somewhat hard time following the math. I do understand, that the  ϕ and ϕj you have used in equations (7) and (8) are functions of ψ's, but it would be really good if you would be more explicit about this. Now it is completely readers burden to fill in the gaps in the details.
>
> We have highlighted the expressions of these quantities and moved them from in-line to having their own line to highlight their importance and for the paper to better readability.
>
> > * The $\lambda_j$ in Equation (Local FedHDP objective) is not explained at all, and is not present in the "global" version above (eq. (9)). Please explain what $\lambda$ is. (I guess it is some type of regularization parameter but why is it only present in clients end and not in global model?)
>
> We have highlighted that this quantity is a regularization parameter to control the closeness of the personalized model to the global model. It is only present on the clients’ end because it’s in their personalized loss function and not in the loss function they use when cooperating in training the global model. Although it only shows up in the personalized objective for clients, it is not in conflict with anything that is related to the global model. In other words, the objective used in training the global model can have its own standalone regularization function without conflicting with the personalization objective function.
>
>
> > * Before eq. (9) you say "... case of regression ... will be presented in the next section". Which Section are you referring to? In 3.2 there is no word about linear regression. Also Section 4 does not go any deeper on the details of this model.
>
> We have made an error, we were referring to the following subsections (now combined in Section 3.1). We have fixed it. We note that federated point estimation is a form of regression where a single point is to be estimated.

---

### Review · Reviewer_RNE8 · 2022-08-11

**Summary Of Contributions:**

The paper studies federated learning in a setting where different users have different privacy budgets. In this setting, each client $k$ can set a value of $\epsilon_k$ and the overall model will have a privacy budget $\max_k \epsilon_k$. Then the authors focus on the private/public data setting.

The authors study the specific problem of federated point estimation with public and private data and propose FedHDP to solve it. They find the rate that allows FedHDP convergence to the optimal value, and study the gap between the expected variance introduced by FedHDP, and the variance of DP-FedAvg.

These results are used as a motivation for a more general Federated learning algorithm that is tested on MNIST and FEMNIST, showing this framework’s global model can outperform baselines. The personalized models performance all have similar results across methods.


**Broader Impact Concerns:**

None.

**Requested Changes:**

In addition to addressing the weaknesses above, how are DP-SGD parameters selected? It is mentioned that FedHDP was tuned, but there is no reference to DP-FedAvg hyperparameters. If DP-SGD hyperparameters are fixed then the comparison is not fair. In either case hyperparameter tuning would break the privacy so I would suggest a different way of presenting the results that would reflect this difficulty. Perhaps graphs of accuracy in terms of varying parameters.

_Minor details_

Clarify add/remove model in the definition of differential privacy.

_Miscellaneous_

Proper use of \citep and cite. For example in page 3, the second contribution reads “[...] introduced by (Li et al. 2021)”
Page 8, in remark “First, The server…” should not be capitalized.


**Strengths And Weaknesses:**

**Strenghts**

- The paper studies a simple problem (point estimation) to derive an algorithm for federated learning with public and private clients. This is a significant problem given that current DP optimization algorithms suffer from poor performance. Yet, in many settings public data is available, thus studying how to combine private and public data, and tradeoffs is an interesting problem for the community.
- The paper quantifies the rate of private users the system can tolerate to learn (asymptotically) the optimal parameter.

**Weaknesses**

- Presentation:
1. In general I found hard to follow where the paper was going, since it starts discussing (general) federated learning with different privacy budgets. Then, the paper goes back to single point estimation, and based on those results presents a heuristic for FL, with private and non-private users. Perhaps just adding a better “organization” section in the introduction, making more clear the specific contributions, and/or adding transitions could help.
2. The notation is very heavy, and sometimes hard to follow.

- Content:
1. The paper proposes heterogeneous DP, defined as a setting where each client controls its privacy budget, but this is misleading since the paper studies a subclass of these problems: learning from private and public data. It is hard to motivate the heterogeneous setting, particularly because interpreting epsilon is an open question for the DP community, as mentioned by the authors.

2. Before stating Lemma 1, the paper states that the result is based on Lemma 11 from Mahdavifar et al. (2018) but I could not find that result in the referenced paper.
3. The algorithm requires tuning hyperparameters which makes the achieved performance non-private.
Improvement over baselines in personalized models is not significant, and in federated learning this metric is typically more important than the global model performance.

- Related work. The paper is missing references to recent work on training from public and private data, some of which state results similar to the point-estimation ones.

https://arxiv.org/pdf/2111.00115.pdf

https://proceedings.mlr.press/v162/amid22a/amid22a.pdf

https://arxiv.org/pdf/2007.03813.pdf

---

> ### Author Response · Authors · 2022-08-26
> **Response to Reviewer RNE8: Part 2**
>
> > 2. Before stating Lemma 1, the paper states that the result is based on Lemma 11 from Mahdavifar et al. (2018) but I could not find that result in the referenced paper.
>
> We have corrected the date in the text. It can be found here: <https://doi.org/10.1109/TSIPN.2017.2749969>. Also, as for the lemma, you can find it in the appendix of the cited paper on page 507.
>
>
>
> > 3. The algorithm requires tuning hyperparameters which makes the achieved performance non-private. Improvement over baselines in personalized models is not significant, and in federated learning this metric is typically more important than the global model performance.
>
> We understand the concern about the hyperparameter tuning. We point out that other works that design DP algorithms by using hyperparameters tuned using non-private baselines also suffer from the same concern here, see <https://arxiv.org/pdf/2110.03620>. We have added a discussion on this part as a part of a remark in Section 5, as well as in the limitations of this work at the end of the paper. It is worth noting that in our experiments, we found that a small enough $r$, between 0.01 and 0.1, provides improved performance compared to the private baselines. As to your concern about the lack of improvement in terms of personalized model performance, we respectfully disagree with such a statement. Our experiments show that there can be a considerable gain in terms of performance on local models at clients compared to the baselines, see the columns dedicated to the performance of the personalized local models in Table 4 for example, or refer to the appendix for extended results. Additionally, for the federated point estimation problem, an improvement over the baselines can be seen in Figure 2.
>
>
> > * Related work. The paper is missing references to recent work on training from public and private data, some of which state results similar to the point-estimation ones. <https://arxiv.org/pdf/2111.00115.pdf>
> <https://proceedings.mlr.press/v162/amid22a/amid22a.pdf>
> <https://arxiv.org/pdf/2007.03813.pdf>
>
> We thank the reviewer for bringing the references to our attention. To address this comment, we have added the references to the related work section in the revised version.
>
> > **Requested Changes:**\
> > In addition to addressing the weaknesses above, how are DP-SGD parameters selected? It is mentioned that FedHDP was tuned, but there is no reference to DP-FedAvg hyperparameters. If DP-SGD hyperparameters are fixed then the comparison is not fair. In either case hyperparameter tuning would break the privacy so I would suggest a different way of presenting the results that would reflect this difficulty. Perhaps graphs of accuracy in terms of varying parameters.
>
> To clarify, we have used the non-private baseline to tune the hyperparameters of the algorithm, such as learning rate, etc., for the specific dataset and model. Once these parameters are tuned, we use them for the remaining algorithms and baselines. We specifically mention tuning the proposed algorithm FedHDP because we have an additional hyperparameter to fine tune which is the ratio hyperparameter $r$, keeping in mind the other hyperparameters are still the same. As we have mentioned in the response to point 3 above, we have addressed this in Section 5 and added it to the limitation section.
>
>
>
> > *Minor details*\
> > Clarify add/remove model in the definition of differential privacy.
>
> To address this comment, we have clarified what is meant by the adjacent inputs in the definition. Please let us know if that is what was requested.
>
> > *Miscellaneous*\
> > Proper use of \citep and cite. For example in page 3, the second contribution reads “[...] introduced by (Li et al. 2021)” Page 8, in remark “First, The server…” should not be capitalized.
>
> We have corrected this in the revised version.

---

> ### Author Response · Authors · 2022-08-26
> **Response to Reviewer RNE8: Part 1**
>
> We appreciate the reviewer’s feedback on the paper and highlighting the insightfulness of the federated point estimation analysis and our solution showing the tradeoffs that arise in such a setup. We include our response to the weaknesses next.
>
> > **Weaknesses:**
> > * Presentation:
> > 1. In general I found hard to follow where the paper was going, since it starts discussing (general) federated learning with different privacy budgets. Then, the paper goes back to single point estimation, and based on those results presents a heuristic for FL, with private and non-private users. Perhaps just adding a better “organization” section in the introduction, making more clear the specific contributions, and/or adding transitions could help.
>
> We thank the reviewer for the suggestions to improve the flow of the paper. Based on the feedback from you and other reviewers, we have now added more details at the end of the introduction around the organization of the paper. We have also significantly revised Section 3 to make clear what is to be expected from this section. We hope that the revisions address your concerns.
>
>
> > 2. The notation is very heavy, and sometimes hard to follow.
>
> We appreciate the concern about the notation. We made an effort to simplify the notation, especially in Section 3, and hence simplified the expressions resulting from such modifications. We also added Table 1, which summarizes the notation that is used throughout this section. We sincerely hope that the revised version clarifies the flow of the ideas. We would also appreciate it if you could let us know whether there are any remaining concerns regarding the presentation.
>
> > * Content:
> > 1. The paper proposes heterogeneous DP, defined as a setting where each client controls its privacy budget, but this is misleading since the paper studies a subclass of these problems: learning from private and public data. It is hard to motivate the heterogeneous setting, particularly because interpreting epsilon is an open question for the DP community, as mentioned by the authors.
>
> We appreciate the reviewer’s comment regarding the setup of the paper We address this issue in two parts.\
> First, we have added a new subsection in the appendix (Section A.6) to extend the theoretical analysis to more than 2 privacy levels, where we provide a proof sketch to derive the optimal values of hyperparameters that arise in FedHDP. The exact expression of the ratio hyperparameters are computed, while regularization parameters for each set of clients are not explicitly computed since those expressions would be quite involved with little insight to be gained. Hence, we show the specific steps needed to derive such expressions and leave out computing the final values of such parameters. We note that the goal of presenting the two privacy level analysis is done so that we can interpret the results and derive insights into the algorithms and potentially utilize such observations to extend the setup to more complex models.\
> Second, as we have mentioned in the paper and you highlighted here, interpreting the values of $(\epsilon,\delta)$ in DP is not straightforward, so in practical settings the casual clients are not expected to have complete knowledge of what the implications of their choices in terms of privacy-utility trade-offs are. As a result, we think that simplifying the choice of privacy for clients as non-private and private is more practical. We added a few sentences at the beginning of Section 3.1.1 to better motivate this choice.

---

### Review · Reviewer_ELsr · 2022-08-12

**Summary Of Contributions:**

The authors propose a FL algorithm that supports heterogeneous DP.

**Requested Changes:**

Please correct the grammar and format issues.

**Strengths And Weaknesses:**

Strength:

- The studied problem, heterogeneous DP, is a very practical challenge in many distributed DP systems, not only FL systems.
- The authors abstract the problem as a single point estimation problem, provide analysis for optimal solution and demonstrate the utility gain with heterogeneous DP.
- Briefly, the algorithm proposed is to aggregate updates from clients with the same privacy level first using DP and then aggregate the aggregated updates from different subsets. Although the idea is straightforward, it seems to work well as illustrated in the evaluation section.

Weakness:

- There are a lot of grammar and format issues. Please take a careful pass to fix them. Some of the issues I noticed are listed below. page 1: "the server has access to a select number of clients". "select" should be "selected". page 2: "In (Avent et al., 2017)," please remove the brackets page 3, the 2nd point in contribution, " introduced by (Li et al., 2021),", please remove brackets Table 2 seems to be horizontally stretched?

---

> ### Author Response · Authors · 2022-08-26
> **Response to Reviewer ELsr**
>
> We appreciate the reviewer’s encouraging comments on the paper and thank them for the positive comments on the novelty of the setup and our approach to solving the problem. Next, we include our response to the weaknesses part.
>
> > There are a lot of grammar and format issues. Please take a careful pass to fix them. Some of the issues I noticed are listed below. page 1: "the server has access to a select number of clients". "select" should be "selected". page 2: "In (Avent et al., 2017)," please remove the brackets page 3, the 2nd point in contribution, " introduced by (Li et al., 2021),", please remove brackets Table 2 seems to be horizontally stretched?
>
> We thank the reviewer for the comment. We have revised the paper and corrected some typos and language issues that we came across. We have also fixed the tables to be aligned with each other. We would just want to clarify that we believe the usage of “select” in this context is correct and would like to keep it as is.

---

### Review · Reviewer_KW6e · 2022-08-15

**Summary Of Contributions:**

The paper considers the problem of heterogeneous privacy for federated learning. Using a Bayesian setup for federated point estimation and linear regression problems, it shows a trade-off between utility and privacy. The full algorithm FedHDP is written, and then experimentally verified.

**Broader Impact Concerns:**

I do not see any broader impact of this work. There is no ethical concerns.

**Requested Changes:**

See the weaknesses part.

**Strengths And Weaknesses:**

Strengths: The problem of heterogeneous privacy level consideration is interesting, and it is good to see a trade-off between the private and non-private clients.

Weaknesses: Having said that I have the following concerns:

Writing: The paper is very poorly written. In particular, the writing in Section 2 needs to be improved a lot. It is very difficult to understand the objective of the paper from this section. The section starts with client indexed by $i$, very quickly moves to $j$, then introduces index $c$ without explaining it properly. After that again new indices are introduced, $p_i$. It is very difficult to follow.

Apart from this, in this section, there are some technical issues. In Definition 1, the privacy is defined with neighboring datasets, that differ in one entry. With that Equation 2 violates the definition, as $D_j$ and $D_e$ vary in $n_i$ data points. Also, how is $A_j(D_e)$ defined where $D_e$ is an empty set. Please explain. Equation 2 makes little sense to me in terms of privacy definition. Clarify this.

The term $\tilde{D}_j$ is not defined properly. The authors use $D$ for both the data set as well as the output of a randomized algorithm. Please clarify.

Also, it is assumed that a client observes the randomized outputs for all other clients? Why is this true? This is not the case in standard Federated Learning, as clients do not broadcast. The clients only get to see the average or a function of other clients output. Please state clearly what I am missing here.

Moreover, the federated point estimation problem is quite restrictive. In Section 3, it is said that all the necessary details are sent to appendix. Why? when there is no page limitations. It is not at all readable without the necessary information from Appendix. For example, equations 7 and 8 need more explanation. Also, define $w_i$ in equation 9. Again, why do clients have access to $\psi_1,\ldots,\psi_N$?

Novelty: The paper has serious organizational issues. The algorithm FedHDP is not written formally before the results in Section 3, which is strange. Moreover, in the technical section, the analysis is only done for point estimation and linear regression. Moreover, the framework just considers 2 cluster cases, with private and non-private clients, despite advertising $\ell$ clients and full generality in Section 2. These things should be clarified upfront. The technical novelty is hence limited. There is no analysis for general cases beyond regression tasks.

On top of this, there are several typos and minor issues throughput the paper. Please correct them.

---

> ### Author Response · Authors · 2022-08-26
> **Response to Reviewer KW6e: Part 2**
>
> > Also, it is assumed that a client observes the randomized outputs for all other clients? Why is this true? This is not the case in standard Federated Learning, as clients do not broadcast. The clients only get to see the average or a function of other clients output. Please state clearly what I am missing here.
>
> Thanks for this clarification comment. In defining the Bayes optimal setup, we are trying to find the optimal quantity that a client can learn subject to having access to a version of the data from all other clients that does not violate their privacy. This is the objective in (Local Bayes objective). We completely agree with the reviewer, as we mention in the paper, that this is impractical; instead in federated learning, the objective of the client becomes a personalization based on some global model that aggregates all clients. This is the objective presented in (Local personalized federated objective). We note that the solution to the federated personalized objective is generally a worse solution than that of the (impractical) Bayes optimal solution. However, in federated point estimation we find that the two solutions coincide, which allows us to devise an algorithm that is Bayes optimal, i.e., there is no other algorithm (be it federated or not) that can lead to clients achieving better performance (measured through estimation variance). We have revised this section to clarify these points.
>
>
> > Moreover, the federated point estimation problem is quite restrictive. In Section 3, it is said that all the necessary details are sent to appendix. Why? when there is no page limitations. It is not at all readable without the necessary information from Appendix. For example, equations 7 and 8 need more explanation. Also, define $w_i$ in equation 9. Again, why do clients have access to $\psi_1,…,\psi_N$.
>
> We have revised the paper to have better readability, especially Section 3 and its discussion on federated point estimation. We have made sure to include all the details needed to reach the goal of such discussion. As to the comment of clients having access to the updates, please refer to the response to the previous .
>
> > Novelty: The paper has serious organizational issues. The algorithm FedHDP is not written formally before the results in Section 3, which is strange. Moreover, in the technical section, the analysis is only done for point estimation and linear regression. Moreover, the framework just considers 2 cluster cases, with private and non-private clients, despite advertising ℓ clients and full generality in Section 2. These things should be clarified upfront. The technical novelty is hence limited. There is no analysis for general cases beyond regression tasks.
>
> We appreciate your feedback on the paper. In Section 3, we had referred to a version of FedHDP (Algorithm 2 prior to revision) that was in the appendix. We have now moved that specific algorithm to Section 3 (Algorithm 1 in the revised paper) and have revised the paper to be more clear about the specific discussion presented in the paper. We have also extended the results to arbitrary number of sets of clients in Appendix A.6 and updated the contribution section to include the contributions and where they are in the paper. We note that analysis of federated learning algorithms in full generality is hard (if not impossible) with current tools. Hence, we believe that, though limited, the analysis provided in this paper sheds some light on how to design differentially private algorithms. For example, we believe that our finding that optimal differentially private federated learning requires personalization even if there is no data heterogeneity or privacy heterogeneity is novel and insightful (Table 2, page 9).
>
>
> > On top of this, there are several typos and minor issues throughout the paper. Please correct them.
>
> We have revised the paper for typos and language errors.

---

> ### Author Response · Authors · 2022-08-26
> **Response to Reviewer KW6e: Part 1**
>
> We appreciate the reviewer’s comments on the paper and its strengths presenting a heterogeneous differential privacy in federated learning. We present our response to the weaknesses next.
>
> > **Weaknesses:**\
> > Writing: The paper is very poorly written. In particular, the writing in Section 2 needs to be improved a lot. It is very difficult to understand the objective of the paper from this section. The section starts with client indexed by i, very quickly moves to j, then introduces index c without explaining it properly. After that again new indices are introduced, pi. It is very difficult to follow.
>
> We have revised the paper to address the clarity of the presentation,  as requested by you and the other reviewers. We hope the revised version is more organized and provides better readability. We have also fixed the few index issues in that section to be consistent with the remainder of the paper. In general, we use the index $j$ to refer to a specific client while we use the index $i$ to refer to a parameter that is related to a set of clients. We wanted to make this differentiation very clear to allow flexibility in the discussion and derivations in the rest of the paper.
>
> > Apart from this, in this section, there are some technical issues. In Definition 1, the privacy is defined with neighboring datasets, that differ in one entry. With that Equation 2 violates the definition, as $D_j$ and $D_e$ vary in $n_i$ data points. Also, how is $A_j(D_e)$ defined where $D_e$ is an empty set. Please explain. Equation 2 makes little sense to me in terms of privacy definition. Clarify this.
>
> In Definition 1, there was no specification of what the entry is. In the original differential privacy setup, this entry was meant to denote a single sample in a dataset. However, we can have this single entry to denote any type of difference between two adjacent inputs as long as the relationship holds for all adjacent inputs. For example, in federated learning with global differential privacy guarantees (client-level DP), the two adjacent datasets are datasets that differe by removing all samples of any single client, see (Geyer et al, 2017; McMahan et al, 2018) for more on client-level DP. Note that these are the DP guarantees we are concerned with in this paper. So, when we define our DP guarantee from a client-level perspective, from the client’s point of view, its neighboring dataset becomes the empty dataset. Therefore, we get equation (2). To clarify this properly, we have added an explanation after the definition of DP in the paper to reflect this idea.
>
> > The term $\tilde{D}_j$ is not defined properly. The authors use $D$ for both the data set as well as the output of a randomized algorithm. Please clarify.
>
> We have replaced $\tilde{D}_j$ with $\boldsymbol{\psi}_j$ to reflect the case that $\boldsymbol{\psi}$ is a privatized statistics about the dataset $\mathbf{D}_j$ and remove any confusion about it.

---

### Decision · Action_Editors · 2022-10-08

**Recommendation:** Reject

**Comment:**

This paper studies a differentially private (DP) federated learning (FL) setting where the clients have different privacy requirements. By studying the point estimation problem (and linear regression) in the federated setting, the authors designed a new weighted averaging algorithm specifically for this setting. The authors demonstrated their method's effectiveness in classification tasks and showed that when the hyperparameter r of FedHDP is appropriately set, the proposed method outperforms the vanilla FedAvg.

While all the reviewers agreed that the paper is of interest to TMLR's audience, they found that the claims made in this paper are not yet fully supported by accurate, convincing and clear evidence. The theoretical results are either too simple or require very strong assumptions. The baseline used in the experimental results are limited, and the authors are encouraged to further investigate the connection to the related problem setting where public data is being utilized. The choice of a critical hyperparameter was also questioned. Lastly, but not least, the reviewers strongly claimed that the presentation of the paper need a significant improvement.

All these factors considered together, we recommend Reject but strongly encourage the authors to address the aforementioned comments to improve the current manuscript.

**Audience:**

The paper proposed a method to satisfy heterogeneous DP requirements in FL systems. The problem is interesting, and the reviewers could see many scenarios where the differing privacy levels might encourage different types of users to participate. The proposed algorithm is practical and has the potential to be used in real-world systems.

**Claims And Evidence:**

**Theory**

While the reviewers believe that the claims (both theoretical and experimental results) are not wrong, the settings examined in this paper are pretty restricted. The point estimation setting might be relevant for some elementary statistics, but reviewers were unsure if federated learning is the proper framework for this case. In the linear regression setting, the authors made strong assumptions that it is not practical.

**Experiments**

[Hyperparameters] The experiments support the proposed framework and authors' claims. However, the reviewers are still unsure how the extra hyperparameter r used by FedHDP would affect the privacy bound. Given that the choice of r significantly affects the performance, the reviewers believe the paper should better discuss this to draw firm conclusions about the observed gain.

[Leveraging public data] This work closely connects with approaches that leverage public data (either directly or indirectly through a pretrained model) to improve the utility of private models. It is reasonable to expect that one could apply some variations of the algorithms (or the ideas) proposed in that setting to the HDP setting. Such comparisons would clarify the distinctions and similarities between these two relevant problem settings.

**Clarity/Presentation**

Many reviewers believe that the paper requires significant rewriting. Even though the authors uploaded a revision that addressed some of the issues, the reviewers strongly believe that the presentation of the paper could still be significantly improved.